# Comprehensive Review of Methodology to Detect Reactive Oxygen Species (ROS) in Mammalian Species and Establish Its Relationship with Antioxidants and Cancer

**DOI:** 10.3390/antiox10010128

**Published:** 2021-01-18

**Authors:** Shivkanya Fuloria, Vetriselvan Subramaniyan, Sundram Karupiah, Usha Kumari, Kathiresan Sathasivam, Dhanalekshmi Unnikrishnan Meenakshi, Yuan Seng Wu, Mahendran Sekar, Nitin Chitranshi, Rishabha Malviya, Kalvatala Sudhakar, Sakshi Bajaj, Neeraj Kumar Fuloria

**Affiliations:** 1Faculty of Pharmacy, AIMST University, Kedah 08100, Malaysia; sundram@aimst.edu.my; 2Faculty of Medicine, Bioscience and Nursing, MAHSA University, Kuala Lumpur 42610, Malaysia; drvetriselvan@mahsa.edu.my (V.S.); sywu@mahsa.edu.my (Y.S.W.); 3Faculty of Medicine, AIMST University, Kedah 08100, Malaysia; usha_harischandran@aimst.edu.my; 4Faculty of Applied Science, AIMST University, Kedah 08100, Malaysia; skathir@aimst.edu.my; 5College of Pharmacy, National University of Science and Technology, Muscat 130, Oman; dhanalekshmi@omc.edu.om; 6Faculty of Pharmacy and Health Sciences, Universiti Kuala Lumpur Royal College of Medicine Perak, Ipoh 30450, Malaysia; mahendransekar@unikl.edu.my; 7Faculty of Medicine and Human Sciences, Maquarie University, North Ryde, NSW 2109, Australia; nitin.chitranshi@mq.edu.au; 8Department of Pharmacy, SMAS, Galgotias University, Greater Noida 203201, India; rishabha.malviya@galgotiasuniversity.edu.in; 9School of Pharmaceutical Sciences (LIT-Pharmacy), Lovely Professional University, Jalandhar 144411, India; sudhakar.20477@lpu.co.in; 10Delhi Institute of Pharmaceutical Science and Research, Pushp Vihar, New Delhi 110017, India; sakshibajaj84@gmail.com

**Keywords:** reactive oxygen species, antioxidants, detection methods, harmful and beneficial effects

## Abstract

Evidence suggests that reactive oxygen species (ROS) mediate tissue homeostasis, cellular signaling, differentiation, and survival. ROS and antioxidants exert both beneficial and harmful effects on cancer. ROS at different concentrations exhibit different functions. This creates necessity to understand the relation between ROS, antioxidants, and cancer, and methods for detection of ROS. This review highlights various sources and types of ROS, their tumorigenic and tumor prevention effects; types of antioxidants, their tumorigenic and tumor prevention effects; and abnormal ROS detoxification in cancer; and methods to measure ROS. We conclude that improving genetic screening methods and bringing higher clarity in determination of enzymatic pathways and scale-up in cancer models profiling, using omics technology, would support in-depth understanding of antioxidant pathways and ROS complexities. Although numerous methods for ROS detection are developing very rapidly, yet further modifications are required to minimize the limitations associated with currently available methods.

## 1. Introduction

Cellular aerobic reactions continuously produce reactive oxygen species (ROS) called reduced oxygen compounds or by-products that perform crucial function [1]. Dysregulation of aerobic metabolism at cellular levels causes elevation of ROS that damages cellular macromolecular components responsible for regulation of signaling and differentiation processes related to cell damage and death [2]. Cancer cells (CC) analysis suggests that ROS in high levels cause damage of cell components (proteins, lipids, and DNA), oncogenicity, genome instability, and tumorigenesis [3,4]. Reports reveal that ROS supports CC proliferation, angiogenesis, metastasis, and survival. The unwanted actions of ROS are regulated via defensive system of cellular antioxidants (AO) [4]. ROS are considered a diverse group of molecules that exert distinctive effects on cell components and over cellular processes, and lead to pro-cancer and anti-cancer effects [2]. CCs elevate ROS production rate via oncogenic mutation, reduction of tumor suppressors, increase in metabolism, and adaptation to hypoxia for hyperactivation of pro-tumorigenic signaling [5]. On the other hand, the excessively high ROS levels may elevate oxidative stress (OS) and cause CC death [6]. To thwart ROS production and preserve redox balance, the CCs elevate their antioxidant capability [7]. The CC scavenges excess ROS up to the level which activates protumorigenic signaling pathways without persuading CC death. In contrast to normal cells, CCs possess altered redox setting, maintaining higher ROS production rate counterbalanced with higher ROS scavenging rate [8]. Such characteristic property of CC as compared to normal cells makes them highly sensitive to ROS level alteration or redox [8]. ROS at different concentrations play different roles in cellular processes; and approach of ROS elimination or ROS production could be an effective cancer therapy. Thus, this review focuses on the relationship between ROS, antioxidants, and cancer; and various methods to detect ROS.

## 2. ROS Sources and Types

ROS contain a minimum of one oxygen [9]. Previous studies have reported various types of ROS, sources, and primary targets given as follows.

### 2.1. Superoxides (O_2_˙^−^)

Reaction of electrons with O_2_ molecules (during electron transport chain reaction of mitochondria) generates various short-lived and moderate reactivity-exhibiting O_2_˙^−^ species. Generally, negative charge over O_2_˙^−^ prevents their diffusion through biological membranes. CCs possess a characteristic property of expressing membrane-associated *β*-nicotinamide adenine dinucleotide phosphate (NADPH) oxidase 1 (NOX1). The extracellular O_2_˙^−^ derived from NOX1 causes autocrine stimulation of proliferation and regulates the intercellular signaling pathways (namely: NO/peroxynitrite and HOCl) to eliminate the transformed cells [10,11]. CCs express catalase (CAT) which protects cells against O_2_˙^−^ signaling that induces apoptosis. The superoxide dismutase (SOD) exhibits a protective role which is related with a CAT-mediated protective effect. Attributed to NOX1, SOD, and CAT co-localization over CC membrane, the inhibition of SOD leads to O_2_˙^−^ based CAT inhibition. This results in apoptotic signaling via the NO/peroxynitrite pathway and reduces the level of H_2_O_2_ (the proliferation stimulator) [12]. Some cells such as neutrophils also generate O_2_˙^−^ species based on the action of NOX which targets iron-sulfur (Fe-S) clusters to release iron. The extracellular O_2_˙^−^ species on reaction with nitric oxide (NO) forms peroxynitrite (ONOO^−^) [13]. This reaction attacks particularly malignant cells with active NADPH oxidase. ONOO^−^ may either react with CO_2_, or if it is generated in close vicinity to proton pumps, it may get protonated to peroxynitrous acid which decomposes into NO_2_ and ˙OH radicals. These are damaging and also can induce apoptosis through lipid peroxidation (LP) [12]. The ONOO^−^ further reacts with proteins (causes oxidation or nitration of amino acids), DNA (causes breaking of double strand), and lipids (causes peroxidation of lipids) [13].

### 2.2. Hydrogen Peroxides (H_2_O_2_) 

The O_2_˙^−^ species dismutate into H_2_O_2_ species by superoxide dismutase 1, 2, and 3 (SOD 1, 2, and 3) enzymatically or non-enzymatically. The H_2_O_2_ species exhibiting long life and moderate reactivity are also generated by Ero1 as result of oxidative protein folding in endoplasmic reticulum [14]. Evidence suggests that selected aquaporin homologues assist in transportation of H_2_O_2_ across membranes [15]. Therefore, these may cause effects distant from their production site. H_2_O_2_ is also a substrate for HOCl synthesis by myeloperoxidase (MPO), dual oxidase (DUOX), or peroxidasins. HOCl undergoes Fenton chemistry much more efficiently than H_2_O_2_ (yielding hydroxyl radicals and chloride). Furthermore, the interaction between HOCl and superoxide anions leads to the formation of hydroxyl radicals. This reaction is important for the elimination of malignant cells [16]. These are the primary ROS that cause protein oxidation. The H_2_O_2_ species at low level of 1–10 nM exert key functions in cellular level signal transduction via protein oxidation (for example protein tyrosine phosphatase (PTP)) and insulin signaling, whereas H_2_O_2_ species at high level of 100 nM causes OS [17]. 

### 2.3. Hydroxy Radicals (˙OH)

The majority of ROS-induced cell damage is attributed to conversion of H_2_O_2_ and O_2_˙^−^ into other ROS. Among all ROS, the ˙OH species are most reactive. These are generated in two ways; firstly by Fenton reaction of H_2_O_2_ with iron (Fe^2+^) and secondly by reduction of Fe^3+^ into Fe^2+^ by O_2_˙^−^ [18].

### 2.4. Lipid Peroxides (LP)

Polyunsaturated fatty acid (PUFA) comprises reactive hydrogens (attributed to carbon–carbon double bonds) that are susceptible for LP which in turn compromises lipid bilayer integrity. The ˙OH induces LP, which generates lipid peroxyl or lipid radicals that further react with PUFA to generate the lipid peroxides. Excess of LP is connected with ferroptosis (iron-dependent cell death) [19]. Figure 1 represents various types of ROS and their sources, and generation in superoxide anions by malignant cells through NADPH activity.

## 3. Protumorigenic Effects of ROS 

Radiation is a well-acknowledged source of ROS. These are correlated with tumor initiating/promoting events. The ROS causes oxidation of nucleic acid bases thereby resulting in damage of DNA; and repairing of such modified nucleic acid bases leads to error that manifests in mutagenesis [20,21]. The cell processes are altered by ROS via their effect over protein functions, and the ROS effect is correlated with the level of protein oxidation. Low oxidation causing promotion of cell signaling is considered reversible (for example, disulfides, sulfinic acid, and sulfenic acid). This also allows rapid modification of protein activity and signaling pathway. However, high oxidation causes terminal oxidation (for example, sulfonic acid) and total loss of protein functioning. Irreversible modification of cysteine may damage protein function, whereas reversible modification may be protective during stress. Modification of proteins assists in adaptation to ROS by triggering the antioxidant Kelch-like ECH-associated protein (KEAP1) or glyceraldehyde 3-phosphate dehydrogenase (GAPDH), and pyruvate kinase (PKM2) programming to assist in metabolism of ROS. However, other endogenous reversible modification such as CoAlation and glutathionylation may also occur to protect protein from terminal oxidation and alter function to promote metabolic networking [22,23]. The ROS effect to initiate and promote tumors involves complex events (that depend upon ROS location, amount, context, and duration).

## 4. Antitumorigenic Effects of ROS 

Excessively high levels of ROS lead to cell cycle arrest, senescence, and CC death [24]. Elimination of CCs in early stages (malignant cells transformation) occurs via intercellular signaling (HOCl or the ˙NO/ONOO^−^ signaling) based induction of selective apoptosis, which involves the ROS and reactive nitrogen species (RNS). Induction of apoptosis by ROS and RNS signaling is initiated through high expression of NOX1 by malignantly transformed cells, which causes high production of extracellular O_2_˙^−^. On one hand, under HOCl pathway, the O_2_˙^−^ are dismutated into H_2_O_2_ either spontaneously or by action of membrane-associated SOD. The formed H_2_O_2_ are converted into HOCl by DUOX coded peroxidase (derived from transformed cells or neighboring non-malignant cells) initiating HOCl signaling [16,25]. The HOCl and NOX1-derived O_2_˙^−^ interaction leads to generation of ˙OH radicals that causes LP. On the other hand, under ˙NO/ONOO^−^ pathway the O_2_˙^−^ may react with ˙NO to form ONOO^−^ which further associates with H^+^ (supplied by membrane associated proton pump) to form ONOOH. This ONOOH decomposes into ˙NO_2_ and ˙OH radical responsible for LP [26,27]. When intracellular AO (GSH) are insufficient to repair damage caused by ˙OH, the caspase-3 and -9 associated apoptosis gets initiated. Such sequential events that lead to apoptosis of transformed cell are referred as intercellular HOCl or ˙NO/ONOO^−^ signal pathway [27,28].

The CC may escape such apoptotic signaling via membrane-associated CAT (MAC) expression that eliminates the H_2_O_2_ near the cell membrane and thereby prevents the synthesis of HOCl and apoptosis inducing HOCl pathway [28,29]. The CAT also interferes ˙NO/ONOO^−^ signal pathway via ˙NO oxidation and ONOO^−^ decomposition [27,28]. Thus, interference of HOCl and ˙NO/ONOO^−^ pathways by MAC leads to survival of tumor. Therefore, inactivation of MAC over CC can reactivate intercellular ROS and RNS based apoptosis [28,30].

Studies suggest that production of singlet delta O_2_ over cell membranes may inactivate MAC selectively, and result in re-activation of intercellular ROS- and RNS-directed apoptosis [31]. Another study suggests that the antitumor effect of cold atmospheric plasma (CAP) may be related to a process involving singlet delta O_2_ [32]. CAP in gas and liquid phase comprises several photons, electrons, ROS, and RNS. CAP-derived ROS and RNS produced in gas phase after getting transferred to liquid medium offers unique biochemical properties attributed to their different multiple interaction potentials, lifetime, and free diffusion path length. Liquid medium treatment with CAP offers plasma-activated medium (PAM) preserving majority of CAP properties. However, PAM comprises long lived species of CAP such as H_2_O_2_, NO_2_^−^, and NO_3_^−^ [33,34]. One study recognized synergy between NO_2_^−^ and H_2_O_2_ as essential for biological potential of CAP. Fact suggests that the role of ONOO^−^ is generated due to interaction between NO_2_^−^ and H_2_O_2_ [35]. CAP and PAM are reported for in vitro and in vivo antitumor activity [34,35]. Clinical investigation suggests CAP potential in tumor therapy is free of severe side effects [36].

Interestingly, in most of the in vitro and in vivo studies on malignant and non-malignant cells, CAP and PAM act selectively towards malignant cells. However, a few studies reported non-selective apoptosis-inducing effects of CAP and PAM [32,37]. This inconsistency can be solved through standardization of dose and composition of CAP and PAM [37]. The in vitro and in vivo response of tumors in different tumor systems reveal that CAP and PAM target the general component of tumor cells. The underlying selective antitumor mechanism of CAP and PAM is yet to be established.

High concentration of aquaporins over tumor cells is considered an important determinant for the selective antitumor mechanism of CAP and PAM [38]. This is because in comparison to non-malignant cells, it allows enhanced influx of CAP- or PAM-derived H_2_O_2_ in tumor cells [39]. Hence, it causes tumor cell apoptosis via Fenton reaction-mediated intracellular effects of H_2_O_2_. In comparison to non-malignant cells, the malignant cells exhibit low cholesterol content. As cholesterol hampers ROS entry into cells, it acts as a determinant for selective action of CAP and PAM towards tumor cells. Both aquaporin and cholesterol models support that in CAP and PAM, the ROS and RNS induces cell death in target cells. Both models exhibited H_2_O_2_ as a major effector of CAP and only effector of PAM. During tumor progression, a phenotype is generated that can be characterized based on enhanced resistance against exogenous H_2_O_2_, however this concept was not considered in both models [40,41]. The tumor progression related to resistance against H_2_O_2_ depends upon MAC expression. The MAC protects tumor cells against H_2_O_2_, oxidizes ˙NO, and decomposes ONOO^−^ [28,29].

In comparison to tumor cells, the non-malignant cells and cells in early stage of tumorigenesis exhibit strong apoptosis when challenged with extracellular H_2_O_2_ or ONOO^−^ [29]. It means pure H_2_O_2_-induced apoptosis in tumor cells is non-selective among non-malignant and tumor cells. Hence, in comparison to tumor cells, the non-malignant cells that are unable to express protective MAC are more susceptible to exogenous H_2_O_2_ [28,29]. As in comparison to non-malignant cells, the tumor cells express less CAT [29], so the protective function of MAC of tumor cells is frequently ignored [42,43].

One study reported a model (derived from analysis of ROS- and RNS-induced apoptosis in non-malignant, transformed, and tumor cells) to describe CAP and PAM selectivity on tumor cells [42,43]. In comparison to non-malignant cells, the outer membrane of tumor cells was characterized based on NOX1, SOD, and MAC expression [44,45]. The study revealed that illuminated photosensitizer-derived ^1^O_2_ inactivated MAC [31]. Inactivation of MAC allows longer survival of H_2_O_2_ or ONOO^−^ to produce a large amount of secondary ^1^O_2_ via reaction between H_2_O_2_ or ONOO^−^ [46]. This further leads to inactivation of MAC and reactivation of apoptosis-inducing ROS pathways. Investigation supports that low level ^1^O_2_ (derived from CAP or via interaction with long lived species of PAM) may interact with tumor cells surface carrying NOX1, SOD, and CAT. Hence, CAP- and PAM-derived species trigger tumor cells’ ability to induce substantial response, with no impact on nonmalignant cells. Investigation reports adequacy of H_2_O_2_ or NOO^−^ (found in CAP and PAM) to produce ^1^O_2_ in sufficient concentrations which allow initial inactivation of few CAT molecules [47]. The reaction is initiated from interaction between H_2_O_2_ or NOO^−^ to form ONOO^−^ [48]. The ONOO^−^ and residual H_2_O_2_ further react to produce primary ^1^O_2_ [49]. This indirect interaction involves various steps such as peroxynitrous acid (ONOOH) decomposition into NO_2_˙ and ˙OH [50]; followed by reaction between ˙OH radicals and H_2_O_2_ to produce hydroperoxyl radicals (HO_2_˙) [51]; and finally, the reaction of ˙NO_2_ and HO_2_˙ produces peroxynitric acid (O_2_NOOH) [52]. The O_2_NOOH deprotonates to peroxynitrate O_2_NOO^−^ which further decomposes to produce ^1^O_2_ [52,53] that inactivates CAT [54]. Thus, H_2_O_2_ and ONOO^−^ derived from free tumor cells massively generate secondary ^1^O_2_, followed by CAT inactivation and consequently HOCl signaling activation. HOCl signaling causes apoptosis induction only when H_2_O_2_ influx into cells deplete GSH (that counteract ˙OH mediated LP effects). Based on ^1^O_2_ mediated MAC inactivation, tumor cells are anticipated to permit aquaporin-mediated H_2_O_2_ influx into the cells [34]. It appears depletion of intracellular GSH is a requirement for apoptosis induction after HOCl signaling based LP. One study suggests that inhibition of aquaporins strongly inhibits PAM mediated apoptosis [55].

The ROS may manifest in cell death by activation of ASK1/p38 and ASK1/JNK pathways [56]. The ASK1 (in inactive state) interacts with TRX (in reduced form). The TRX oxidation by H_2_O_2_ leads to dissociation and activation of ASK1; this triggers anti-apoptotic factor suppression via activation of MKK3/MKK6/p38 and MKK4/MKK7/JNK pathways [56,57]. Reports suggest deactivation of mutation in JNK and p38 pathways in tumors, which suggests that these pathways manifest in CC death [58,59]. The p38 MAPK pathway is described to suppress the tumor and inhibit the Hras malignant conversion [60]. Facts suggest that ROS-mediated activation of p38 and JNK pathways may thwart CC growth, mitosis, and stimulate cell cycle arrest [61,62]. When the CC detaches from the extra-cellular matrix, it causes invasion of basal membrane, thereby increasing the ROS level and decreasing the reduced GSH pool [63]. Investigations report that circulating CCs are not able to proliferate and survive in an oxidized blood environment. Therefore, ROS in excess concentration may prevent distant metastasis [64]. To prevent cell death while metastasis and boost anchorage independent growth (AIG), the CC undergoes metabolic transformation by enhancing their AO property [64,65]. Cytosolic IDH1 dependent carboxylation stimulates AIG by reducing mROS concentration and enhancing the mitochondrial NADPH profusion [65]. Therefore, alteration of ROS-alleviating pathways can be a therapeutic approach to suppress the CC metastasis and proliferation [66].

## 5. Antioxidants (AOs)

The term antioxidant refers to any substance that, in low concentration, delays or prevents the oxidation of substrate [67]. The AO can be endogenous or exogenous.

### 5.1. Endogenous Antioxidants (EnAOs)

Endogenous antioxidants (EnAO) are the enzymes or cofactors that eliminate ROS. SOD, CAT, and glutathione peroxidase (GPx) are the three enzymatic systems that play an important role in the EnAO property of biological systems against free radicals. SOD catalytically converts O_2_˙^−^ (produced via metabolic reactions) into H_2_O_2_ and molecular oxygen (O_2_). Accumulation of H_2_O_2_ is toxic for human body tissues or cells and in presence of Fe^2+^, this H_2_O_2_ is converted into ˙OH via Fenton reaction [68]. To prevent this deleterious phenomenon, CAT (abundant in peroxisomes) converts H_2_O_2_ into H_2_O and O_2_, thereby curtailing the damage by free radicals. As catalase is absent in mitochondria, conversion of H_2_O_2_ into water and lipid peroxides into alcohols is done by GPx. Such combined protective system is stated as first line antioxidant defense system [68]. In mammals, the SOD family of metalloenzymes comprises three members, namely: SOD1 (Cu/ZnSOD), SOD2 (MnSOD), and SOD3 (ecSOD). The encoding of Cu/ZnSOD occurs by mapping of SOD1 gene with chromosome 21, MnSOD occurs by mapping of SOD2 gene with chromosome 6, and eukaryotic extracellular SOD CuZn SOD occurs by mapping of SOD3 gene with chromosome 4 [69]. SOD regulatory function in growth, metabolism, and response to oxidative stress is also crucial in cancer development and survival [70]. CAT (the AO present in all living tissues) is a tetrameric protein comprising four similar subunits and each polypeptide subunit comprises single ferriprotoporphyrin. Encoding of CAT occurs by mapping of *ctt1* gene with chromosome 11. CAT utilizes iron/manganese as cofactor and catalyzes degradation or reduction of H_2_O_2_ to H_2_O and O_2,_ thereby completing the SOD-initiated detoxification. At lower concentration, H_2_O_2_ regulates various processes such as cell proliferation signaling, cell death, mitochondrial functioning, and thiol-redox balance; whereas at higher concentration, H_2_O_2_ is detrimental for cells. Therefore, CAT ability to limit H_2_O_2_ level in cells justifies its importance as AO defensive system [68]. Any deficiency of CAT is linked with various diseases. A study reported oxidative DNA damage and consequent cancer susceptibility in individuals with altered gene expression/activity in CAT [71].

The metabolic cofactor glutathione (GSH) is considered the most abundant endogenous antioxidant (discovered more than 10 decades ago) that plays an important role in detoxification of reactions in CCs [72]. This is a tripeptide that is biosynthesized in two key steps. Firstly, glutamate and cysteine undergo ligation in presence of a glutamate cysteine ligase catalytic (GCLC) subunit to produce a dipeptide. Secondly, the formed dipeptide is converted into tripeptide (GSH) by GSH synthetase (GSS) that incorporates glycine into the structure. In GSH synthesis, the cysteine, glutamate, and glycine are considered rate-limiting metabolites [73]. The enzymes GSH S-transferase (GST) and GSH peroxidase (GPX) utilize GSH as a cofactor for the elimination of ROS. The GST and GPX comprise several families and isoforms but the exact target for each is still unclear [74]. The sulfaredoxin (SRX) and thioredoxin (TXN) are another set of antioxidant systems that reproduce peroxiredoxins (PRDX). PRDX enzymes exhibit catalytic action against H_2_O_2_ [75]. TXN are less abundant small antioxidant proteins [76]. On one hand, TXN reduces the disulfide bonds of PRDX, and on the other hand SRX reduces the PRDX that over-oxidizes to sulfinic acid. The PRDX and TXN proteins are localized on mitochondria or cytoplasm, but the relative importance of each and crosstalk among them is still unclear [75]. ROS detoxification via GSH and TXN produces an oxidized form of these AO that are further reproduced by GSH reductase (GR) and TXN reductase 1 and 2 (TXNRD) using NADPH as electron donors for consequent reactions [77]. However, such pathways are complementary, and the redundancy amongst GSH and TXN occurs in normal as well as malignant tissues [78]. Additionally, OS promotes enzymatic expression in both systems of GSH and TXN, which suggest that they work jointly to buffer OS [78]. EnAOs exhibit their importance in tumors based on the stage of tumorigenesis. EnAOs prevent tumor initiation via prevention of ROS-induced oxidation and damage of DNA. Although several studies on carcinogens-induced tumor models support EnAO participation in carcinogens detoxification. However direct role of EnAO in prevention of ROS induced tumor initiation is still unclear.

#### 5.1.1. Tumor Prevention by EnAOs

ROS detoxification by EnAOs reveals their ability to protect from deleterious and oncogenic effects. Numerous isoforms of GST are reported to prevent tumor initiation in skin, the liver, and the colon on exposure to carcinogens or loss of tumor suppressors [79,80]. The GPX enzyme system provides protection against carcinogens and ROS-induced tumor initiation in various models. GPX3 enzyme is reported to suppress the initiation of tumors in the colon cancer mouse model [81]. Mice with low SOD2 expression or in combination with low GPX1 enzyme exhibit high DNA damage and incidence of tumors [82,83]. One study [84] suggests SOD mimetics localized into mitochondria inhibits the proliferation of CC and tumor growth.

The TXN system is also known to exhibit tumor suppressive abilities. Loss of PRDX1 causes DNA damage and high incidence of tumors in old mice. PRDX1 reduces PTEN (phosphatase and tensin homologue deleted on chromosome 10) to stimulate its phosphatase activity against AKT (protein kinase B) and thereby inhibits the growth of CCs. Apart from this, PRDX6 loss hastens human papillomavirus type 8 (HPV8)-induced carcinogenesis of skin [85,86]. Moreover, TXN and GSH in combination exhibit tumor prevention in mice with combined loss of liver-specific TXNRD1 and GR; having high sensitivity towards carcinogen-induced liver malignancy [87].

A high amount of oxidized DNA promotes initiation of tumors in mice, and OGG1 (the enzyme that repairs 8-oxo-deoxy-guanine in DNA) loss causes spontaneous lung tumors in mice in absence of carcinogenic therapy [88]. In vivo genetic study of Gpx2^−/−^ mice exhibited protection against azoxy methane induced colorectal tumorigenesis during early stages [89]. Hence, based on these facts, it is evident that EnAOs exhibit an important function in the prevention of tumors.

There are few in vivo studies that showed EnAOs promoting tumor initiation. Srx^−/−^ mice exhibited few smaller urethane-induced lung tumors and DMBA/TPA-induced skin tumors [90,91]. Some investigations on one hand suggest evidence for EnAOs preventing cancer initiation, whereas on the other hand, some studies suggest that EnAOs promote tumor initiation. In a nutshell, there is a need to study the direct relation of ROS and protein antioxidant function. In addition, further studies are required to ascertain whether tumor suppressing antioxidant systems could be upregulated to prevent cancer without inducing promotion of cancer.

#### 5.1.2. Tumor Progression by EnAOs

Generally, during cellular transformation the cellular processes such as mitochondrial metabolism and translation of proteins are upregulated which causes an increase in ROS production, which necessitates the antioxidants for redox balance [92]. Investigation reveals GSH to prevent DNA damage and retain protein homeostasis in tumors [93]. In case of insufficient GSH production, the tumor cells progress to advance and aggressive malignancy, while progression of tumor GPX and GST enzyme participate in downstream use of GSH. Generally, GST enzymes metabolize chemotherapy (example: cisplatin) and triggers oncogene signaling protein (example: Akt) [94,95]. It needs GPX system to buffer ROS production during tumor progression. For example, GPX4 inhibits LP and ferroptosis [96].

The ferroptosis involves several processes, such as GSH independent pathway which uses AO cofactor ubiquinone (CoQ10) [97]. The therapy-resistant CC which underwent epithelial-mesenchymal transition (EMT) is highly sensitive towards ferroptosis [98]. The TXN components, for example TXN and TXNRD1, are reported to promote tumor growth [99]. In cancer tissues, a high expression of PRDX1 and 4 supports survival of tumors [100] and over-expression of PRDX6 further accelerates tumor progression [101].

O_2_˙^−^ metabolism contributes a key role in the progression of tumors. SOD1 inhibition via copper chelators blocks tumorigenesis in the lung; also, copper chelation exhibits SOD1-independent anti-tumorigenic potential. Targeting of the TXN or SOD1 system impairs lungs CC survival on exposure of O_2_˙^−^ [102,103]. Hence, based on the evidence, it can be established that EnAOs have the ability to support tumor progression. Substantial redundancy between AO systems is left as a challenge for cancer treatments and extensive work is required to understand compensating mechanism between various protein components in such systems [99]. The NADPH (that reproduces EnAO in GSH and TXN system) should be reproduced from NADP^+^. Reproduction of NADPH is stimulated by various metabolic processes, noticeably via pentose phosphate pathway (PPP) and one carbon metabolism [104]. The glucose-6-phosphate dehydrogenase (G6PD) produces NADPH in first step of PPP [105]. Various studies revealed the significance of NADPH reproduction and reduction capacity against tumorigenesis [106]. One study revealed G6PD-deficient patients to exhibit low risk for colorectal cancer [107]. However, it is difficult to determine the G6PD deficiency contribution against risk for cancer as the role of G6PD goes beyond GSH and TXN system regeneration. Therefore, some more studies are needed for in depth understanding of the subcellular relationship between GSH/TXN, NADPH, G6PD, and ROS in tumorigenesis.

### 5.2. Exogenous Antioxidants (ExAOs)

Exogenous antioxidants (ExAOs) are a large group of molecules, that can be divided into three subgroups, namely: polyphenols, vitamins and derivatives, and AO minerals [108]. Polyphenols (the most abundant natural AO) are categorized as flavonoids and phenolic acids. Flavonoids are further categorized as catechins, flavones, flavanones, isoflavones, flavonols, and anthocyanins [109]. Polyphenols are found in plant products such as fruits, vegetables, juices, tea, wine, etc. [110]. One study suggests that polyphenols confer protection against cancer [111]. Among vitamins and derivatives, vitamin C, E, and K, and carotenoids are considered as the most important AOs. Carotenoids are the group of pigments present in several fruits and vegetables. Among 600 types of carotenoids, *β*-carotene and lycopene exhibit high antioxidant properties [111]. Hepatic catabolism of *β*-carotene leads to vitamin retinol that may neutralize peroxyl radicals [112]. Vitamin C with its electron-donating property prevents accumulation of free radicals such as O_2_˙^−^, H_2_O_2_, ˙OH, ^1^O_2_, and RNS [113]. The family of vitamin E includes highly lipophilic molecules of tocotrienols and tocopherols which exert AO activity attributed to their capability to join biological membranes and protect against LP [114]. Though vitamin K is not a classic AO, a study however suggests its ability to reduce the GSH depletion caused by OS [115].

Among the mineral group of AOs, selenium acts as cofactor of AO enzymes such as GPX and thioredoxin reductase [116]. Selenium acts as part of SOD and inhibits NADPG oxidase (enzyme which catalyzes conversion of oxygen to single oxygen radical [108]. Moreover, zinc is reported to prevent LP and protect cell membranes [117].

Generally, ExAOs are used in the treatment of in vivo cancer models and in the determination of the causative role of intracellular ROS in different tumorigenic processes. A study of treatment of patients with neck and head cancer with radiotherapy and high doses of vitamin C and E revealed improvement in adverse effects [118]. Vitamin consumption linked to improvement of adverse effects of chemotherapy and radiotherapy is also supported by other studies [119,120]. Curcumin in combination with radiotherapy is known to offer synergistic antitumor effect in prostate, breast, colorectal, and ovarian cancer [121]. The head and neck squamous cell carcinoma (HNSCC) cell lines study of curcumin and radiotherapy combination revealed curcumin to offer synergistic antitumor effect [122]. The epigallocatechin-3-gallate (catechin) is another radiosensitizer that exhibits synergistic antitumor effects on multiple myeloma (IM-9), glioblastoma multiforme, leukemia (K-562), and cancer cervix (HeLa) cells [123]. A recent study revealed that catechin improves the prognosis of breast cancer patients undergoing radio treatment [124]. The fact that melatonin slows the repair enzymes saturation suggests its ability to repair the OS-induced damage and allow radiotherapy use in high doses. This makes melatonin a protective agent during radiotherapy. Due to its non-toxicity (up to 250 mg/kg), melatonin can be used in very high doses. Reports suggest melatonin to be effective when administered in low doses (0.1 mg/kg/day) to mice for 15 days [125,126]. However, investigation of suitable dose of melatonin for humans with radiotherapy is yet to be established.

Chemotherapy involves several agents that may cause OS-induced cell death in two ways. One is by direct interruption of redox signaling and ROS scavenging; and second is by indirect reduction of intracellular AO to deactivate cellular defense. There are several chemotherapeutic agents that induce OS. Some new molecules include meroxest (synthetic mero-sesquiterpene derivative of *trans*-communic acid that is obtained in plenty from *Cupressus sempervirens*), and Jadomycin (synthesized by *Streptomyces venezuelae*) [127,128]. There are several other agents that constitute current therapeutic repertory such as bleomycin, oxaliplatin, bortezomib, capecitabine, gemcitabine, celecoxib, arsenic trioxide, and cyclophosphamide [111]. However, rarely has any study reported the interaction of AO and chemotherapeutic agents’ antitumor activity. Anthracyclines, the antitumor antibiotics, are linked with OS and increased ROS levels; and are suggested to mediate apoptosis through activation of caspases 3 and 9 [129]. In solid breast and prostate tumors, doxorubicin is known to exert its antitumor activity by inhibiting topoisomerase II and generating ROS which leads to DNA damage and cell death by apoptosis [130]. The increase in ROS plays an important role in cardiotoxicity caused by doxorubicin [131]. One study suggests that administration of AO may thwart the toxicity of doxorubicin in cardiomyoblasts. For instance, vitamin E offers cardio protection against chronic cardiotoxicity, not against chronic cardiomyopathy [132]. Doxorubicin is known to reduce apoptosis in cardiomyocytes and OS in the heart failure model of Japanese white rabbits that were prior treated with *N*-acetyl cysteine (NAC) [133]. A study on the effect of vitamin C on doxorubicin-induced cytotoxicity revealed that vitamin C in high doses offers greater resistance to treatment in myelogenous leukemia (K562) and lymphoma (RL) cell lines [134].

The combined treatment of doxorubicin, cyclophosphamide, and 5-fluorouracil reduces the AO levels which causes LP in cell membranes [135]. A clinical study over efficacy of *Uncaria tomentosa* (UT) in stage II patients of invasive breast ductal carcinoma treated with cyclophosphamide revealed that patients receiving cyclophosphamide with 30 mg/day of UT extract experienced reduction in adverse effects such as neutropenia without affecting the drug’s antitumor activity [136]. Furthermore, tannins (polyphenol) when administered with doxorubicin lowered the cardiotoxicity caused by doxorubicin, without affecting the drug’s antitumor activity. A study suggested that administration of epigallocatechin-3-gallate with doxorubicin offers synergistic effects in a hepatocellular carcinoma (HCC) chemo-resistant model. The study revealed that mice administered with epigallocatechin-3-gallate and doxorubicin showed lower growth rate of liver tumors in comparison to mice administered with only doxorubicin [137]. Other studies also supported the potential of epigallocatechin-3-gallate as an adjuvant with cisplatin and 5-fluoro uracil in cancer treatment [138,139]. Taxane (Paclitaxel) is another class of anticancer cytotoxics that is used in the treatment of various tumors. Taxanes are known to induce ROS and modify the permeability of H_2_O_2_-generating mitochondrial membrane. One clinical study revealed reduction of GSH level in blood samples of paclitaxel treated patients, which infers reduction of AO potential of cells [140]. Docetaxel, the derivative of paclitaxel, is a first line drug for the treatment of prostate and other cancers [141]. Docetaxel is known to induce OS via activation of caspase 3 [142]. One of the studies demonstrated pro-oxidant action of docetaxel on MDA-231 and MCF-7, the breast tumor cells [143].

Cisplatin (a heavy metal) is widely applied in the treatment of solid tumors of lymphoma, lungs, testes, and ovary [144]. It is known to produce intense OS and offer numerous side effects attributed to its toxicity [145]. It is associated with expression of antiapoptotic Bax proteins, p53 (tumor suppressor gene), p21 protein (cell cycle regulator), and cleavage of caspases 3 and 9 and PARP [146]. Quercetin is remarkably reported as an adjuvant in cancer treatment with cisplatin. Recent study of cisplatin treatment in ovarian tumor cells (SKOV3 and C13*) revealed that administration of quercetin in high concentrations (40 μM–100 μM) exhibits a proapoptotic effect, and in low concentration (5 μM–30 μM) reduces the ROS-induced damage [145]. The reduction in damage was attributed to increase in SOD, which attenuated the antineoplastic effect of cisplatin. The interaction of quercetin was also investigated with 5-fluoro uracil, taxol, and pirarubicin (drugs used for ovarian cancer treatment), and similar results were obtained [145]. One of the studies supported that AO assists in lowering of tumor cells progression. A study revealed that tumor cells of colon (COLO-205-GFP) in mice on treatment with cisplatin and high-dose supplementation of vitamins A, E, and selenium (five times more than standard diet) along with fish oil exhibited low growth in comparison to the control tumor [147]. One of the clinical studies highlighted the vitamins’ effect over quality of life (QOL) of cervical cancer patients undergoing cisplatin treatment. In the study, a combination of chemotherapy, radiation, and cisplatin was used. In parallel, patients were administered with vitamin C, vitamin E, and *β*-carotene. The pretrial analysis of patients aged between 29 and 73 exhibited lower AO levels (except for vitamin C and zinc) than recommended. Results showed that females taking supplements during treatment exhibited less oxidation damage, improved muscle strength, and less fatigue (compare to female who did not). During the study, the supplement dose comprised recommended daily dose [148]. Curcumin, the radio-sensitizing AO, was also investigated for its role as an adjuvant treatment with cisplatin. Several studies supported curcumin enhancing the cytotoxic activity of cisplatin against liver tumor cells (HA22T/VGH) [135,149] and HNSCC tumor cells (CAL27 and UMSCC) [121].

Various studies evaluated NAC effects during cisplatin treatment. Studies highlight that NAC has the ability to reverse the cisplatin cytotoxicity and proapoptotic effects in human ovarian carcinoma cells (SKOV3), human glioblastoma cells (U87), and rat fibroblasts (Rat1). Intriguingly, administration of NAC before or after 1 h of application of cisplatin may block its proapoptotic effect. Whereas administration of NAC after 8 h of cisplatin application caused no change in the proapoptotic effects [146]. In vivo study highlights the otoprotective property of NAC when administered up to 4 h after the cisplatin application [150]. NAC is reported to enhance the protection against cisplatin induced renal damage when administered intra-arterially [151]. Therefore, the timing and route of administration of AO is an important factor to be considered in cancer treatment. Apart from NAC, other AOs such as lycopene were also evaluated for their ability to reduce the cisplatin toxicity. One study revealed lycopene ability to reduce the cisplatin-induced renal toxicity [152]. NAC was also evaluated for its potential in cancer patients undergoing chemotherapy and radiotherapy. One of the clinical studies on 40 children with acute lymphoblastic leukemia was done to determine effect of oral administration of NAC and vitamin E (400 IU/day) to counter the toxicity due to chemotherapy (doxorubicin, 6-mercaptopurine cyclophosphamide, vincristine, and cytosine arabinoside) and cranial irradiation. The blood sample analysis for malondialdehyde, GPx, TNF-*α*, and liver enzyme levels revealed that children administered with NCA exhibited reduced incidence for toxic hepatitis and less need for blood and platelet transfusions [153].

As melatonin exerts high AO activity via different mechanisms, it is known to be used as an adjuvant in chemotherapy of different type of cancers. A study evaluated effect of oral administration of melatonin (20 mg/day) in non-small-cell lung carcinoma patients undergoing cisplatin and gemcitabine treatment or cisplatin and etoposide treatment; or gastrointestinal cancer patients undergoing oxaliplatin and 5-fluoro uracil treatment. The study revealed that patients receiving melatonin exhibited an enhanced tumor regression rate and survival rate in both cases [154]. The melatonin in reduction of chemotherapy-induced toxicity (MIRCIT) study revealed that patients with advanced non-small-cell lung carcinoma when administered with melatonin combined with chemotherapy did not improved the survival rate although side effects were lowered [155].

Studies recommend NAC as an extensively used AO. Mice treatment with NAC impairs p53 (tumor suppression protein) null lymphoma and growth of lung tumors via prevention of DNA oxidation and related mutagenic outcomes [156]. Figure 2 represents various types of endogenous (EnAO) and exogenous (ExAO) antioxidants.

NAC is reported to inhibit hypoxia inducible factor 1-alpha (Hif1a) stabilization and perturb the hepato-cellular xeno-graft tumor [157]. Despite preclinical data on NAC to aid clinical trials, no advantage was observed in the cancer patients. Several studies suggest that NAC promotes initiation, progression, and metastasis in mouse model of melanoma and lungs cancer [158,159]. Yet, the mechanics of NAC effect on cells redox status is unclear. As NAC contributes to synthesis of GSH synthesis, its AO function could be attributed to hydrogen sulfide formation and protein persulfidation [160].

Vitamin E (α-tocopherol) is recognized for its antitumor potential [161], but advent of beta-carotene to increase cancer incidence questioned the efficacy of vitamin E in cancer. Therefore, a multicenter clinical study was done over vitamin E to prevent prostate cancer, but study was stopped as vitamin E exhibited higher prostate cancer rate [162,163]. Evidence on one hand suggests that vitamin E promotes lung tumors and melanoma progression, and on the other hand it grows and directly prevents lipid oxidation and ferroptosis [164,165,166].

Generally, ExAO understanding becomes more complex when these molecules oxidize themself or produce ExAO independent effects. A recent study suggests that vitamin C (ascorbic acid) can autoxidize to dehydroascorbate (DHA) and may further increase the OS in cells [167]. Apart from that, vitamin C is also reported to negatively regulate the functioning of hematopoietic stem cell (HSC) by promoting ten-eleven translocation-2 (TET2) activity [168]; and NAC is reported to produce hydrogen sulfide (H_2_S) that may influence several metabolic and signaling pathways [169]. Hence, utmost care must be taken while using ExAOs to examine and interpret the impact of intracellular ROS over tumors.

## 6. Unusual Detoxification of ROS in Cancer

AO role in tumor promotion is based on evidence of abnormal activation of Nuclear factor erythroid 2-related factor 2 (NRF2). The NRF2 is an AO transcription factor that is involved in various forms of cancer. In basal condition, the level of NRF2 is restricted, which is attributed to their connection with KEAP1 that targets NRF2 in proteasomal degradation [170]. Cell exposure to OS causes modification of cysteine residues of KEAP1 that manifests in impaired NRF2 ubiquitination and deposition of NRF2. NRF2 fosters transcription of several genes in the AO system, including genes of GSH and TXN AO pathway [171]. NRF2 plays a complex role in various phases of cancer (Figure 3).

NRF2 deposition in cancer reveals that higher AO defensive system aids are involved in the tumorigenic process. In cancer, NRF2 accumulation involves several mechanisms. In cancer (especially of lungs), the mutation of NRF2 and KEAP1 causes disruption of appropriate NRF2 degradation. Studies on lung cancer provide evidence of skipping of NRF2 exon causing elimination of KEAP1 binding domain, and oncogene-guided transcription to raise the NRF2 level [171,172]. The accumulation of NRF2 is also based on inactivation of KEAP1 that is manifested as a result of promoter methylation, sequestration (p62 protein assisted) [173], and modification mediated by oncometabolites methyl glyoxal (MGO) and fumarate [174]. There are numerous studies which have explored the role of NRF2 during tumorigenesis. Investigations of NRF2 knockout mice exhibited NRF2′s role in the incidence and growth of oncogene-guided lung tumors and p62-assisted pancreatic tumorigenesis [172,175,176]. The elimination of KEAP1 further raises the tumor burden in models of liver tumor driven by Myc (transcription factor that is associated with proliferation of hepatocyte during regeneration of liver), and lung tumor by protein Kras^G12D^ (K-Ras protein that mediates RAS/MAPK pathway)/loss of PTEN or p53 [177,178]. In such models, activation of NRF2 was related with reduced ROS levels, oxidation-based DNA damage, and metabolic processes activation. Multiple metabolic pathways for AO defense and proliferation processes (including PPP and serine biogenesis) are regulated by NRF2 and reported to play the major role in AO processes in particular tumor progressive phases [178,179]. The activation of NRF2 leading to change in reactivity of cysteine reveals that NRF2 effects on metabolism are not limited to direct transcription targets [180]. The ROS impact on metastasis is complex and conflicting [181]. Studies suggest cell separation from the extracellular matrix (ECM) stimulates OS which restricts endurance in circulation and AO exhibits protection and promotion of metastasis [182]. NRF2 impact on metastasis is complex, such as NRF2 activation in Kras^G12D^, p53^flox/flox^ lung tumor indirectly promoting stability of BACH1 (transcription factor) through catabolism of heme, that promotes metastasis through BACH1 transcriptional target (effect that can be recapped with AO treatment [182,183]. In pancreatic cancer mouse model guided by Kras^G12D^ and p53 loss-of-function, the ROS stimulated by elimination of TP53-stimulated glycolysis and TP53-induced glycolysis and apoptosis regulator (TIGAR) or NRF2 increased lung metastasis [184]. In the pancreatic cancer model, the ROS could not affect BACH1. The missing expression of DUSP6 raised the activity of ERK and EMT. A study on a pancreatic tumor model explored the effect of kelch-Like ECH-associated protein 1 (KEAP1) deletion in relation to Kras^G12D^ and p53^R172H^ that recapitulated the lung tumor genetics. The deletion of KEAP1 pancreatic tumor model leads to pancreatic atrophy [185]. The NRF2 impact over metastasis depends upon context, which is affected by nature of tissue, ROS-dependent and independent impacts, and dosage of NRF2. The impacts and extent of loss of function of KEAP1 also depends upon context in same tissue and genetics. In one of the models, elimination of KEAP1 was not chosen for Kras^G12D^ or p53^flox/flox^; in fact the LKB1^flox/flox^ model was contrasted with other tumor suppressors [186]. While heterozygous KEAP1^R554Q^ loss-of-function mutant expression enhanced the size of tumor similar to the deletion study of Kras^G12D^; the p53^flox/flox^ model homozygous KEAP1^R554Q^ expression antagonized the formation of the tumor [187]. For an in-depth insight into whether KEAP1 mutation or level of NRF2 activation have a distinct effect on the growth and progression of tumor, however, additional studies are needed for further understanding.

## 7. Methods to Detect ROS

Studies determined the role of ROS as a messenger in cell proliferation, differentiation, survival, apoptosis, and death. ROS at different levels exhibit different functions. For example, ROS in low concentrations induce mitosis and proliferation; in moderate concentration prevents cell cycle; whereas in higher concentration, may activate apoptosis [21,188]. Hence to assess both disease status and health-enhancing effects of antioxidants in humans, it is very important to measure ROS concentration. There are several methods available to measure the concentration of ROS, such as spectrometry, spectrophotometry, chemi-luminescent and fluorescent probes, chromatography, and genetic encoding based fluorescent protein assay [189]. The following sections explain common and newly developed methods for ROS detection, although for reliability of results more than one method is recommended.

### 7.1. Fluorescent Chemicals-Based ROS Detection

The fluorescent chemicals-based detection methods involve oxidant sensitive fluorescent probes that are non-fluorescent prior to oxidation by ROS. The fluorescent probes-based detection methods include dihydroethidium, dichlorodihydro fluorescein, and Amplex Red (impermeable to cells) as most extensively used candidates. Cell permeable methods assess the oxidative state of cell compartments and determine the production of radicals during stimulus [190]. Generally, such probes undergo oxidation via mechanism of one-electron free radical, which yields probe radical intermediate and leads to fluorescent products.

#### 7.1.1. Dihydroethidium (DHE)

Dihydroethidium (DHE) permeates cells and undergoes oxidation in two ways. One is oxidation by O_2_˙^−^ to produce red fluorescent 2-hydroxyethidium (2-OH-E⁺). Secondly, there is oxidation by other oxidants such as ˙OH, H_2_O_2_, or OONO^−^ to produce non-specific red fluorescent ethidium (E^+^). Generally, both 2-OH-E^+^ and E^+^ may be determined through fluorescent chemical methods. In an intracellular experiment, DHE is a choice for ROS determination. Based on disparity among DHE fluorescent products (the 2-OH-E^+^ and E^+^) and O_2_˙^−^concentration, the DHE alone is not sufficient for quantification of O_2_˙^−^ [190]. The 2-OH-E^+^ and E^+^ can be better identified, distinguished, and confirmed using analytical techniques of HPLC and LC-MS [191]. Stoichiometry of reaction between O_2_˙^−^ and DHE is not the same in different cells and tissues; also in addition, the production rate of 2-OH-E^+^ does not always correspond to O_2_˙^−^ [192]. DHE shortcomings in reporting the level of O_2_˙^−^ indicates that further work is needed to determine the correct mechanism and accurate stoichiometry of reaction. O_2_˙^−^ detection can be done using fluorescent dyes designed for sub-cellular organelles. Mito-SOX (conjugate of triphenyl phosphonium with DHE), the mitochondrial O_2_˙^−^ indicator, is a cationic probe that features like DHE. Mito-SOX is reported for its application in the study of fenretinide-induced apoptosis of neuroblastoma, for the detection of mitochondrial O_2_˙^−^. The study thereby revealed the significance of mitochondrial O_2_˙^−^ in inhibiting the respiratory chain and cytotoxicity [193]. The generation of fluorescence from Mito-SOX is considered as complex. This is based on one of the examples, that 2-OH-Mito-E+ can be generated by O_2_˙^−^ and Mito-E^+^ can be produced by cytochrome c, H_2_O_2_, and peroxidase, which is necessary for assistant analysis techniques [194]. Although several studies report use of Mito-SOX and DHE to detect of O_2_˙^−^ in biological systems, interference by other oxidizing radicals prevents them from quantifying O_2_˙^−^ with accuracy [195]. Hence, for confirmation of O_2_˙^−^ presence, the separation and identification of 2-OH-E+ by chromatography becomes necessary.

#### 7.1.2. Dichlorodihydro Fluorescein Diacetate (DCFHDA)

Dichlorodihydro fluorescein diacetate (DCFHDA) allows direct measurement of intracellular redox states. This enters the cell and undergoes hydrolysis into membrane-impermeable DCFH (by esterase), that diffuses inside and undergoes oxidation into fluorescent DCF (by nonspecific oxygen radicals) via non-fluorescent intermediate free radical DCFH˙^−^. Investigation correlated fluorescence intensity of DCF with the OS levels in cells, revealing the reliability of DCFHDA in determining the redox level [196]. The DCFHDA deacetylation into DCFH and its accrual in cell trap determines the application of DCFHDA in detection of ROS level. Different cells exhibit different permeability towards DCFHDA probe that may lead to variation in final equilibrium concentration inside and outside the cell. As a result, esterase activity may vary among cells. Therefore, DCFHDA/DCFH behavior in specific cells must be investigated before their application. The DCFH assists in monitoring of ROS and other oxidants in cell free system [197]. The DCFH can be obtained by treatment of DCFHDA with esterase (for example: acetyl esterase) or strong alkali (for example: NaOH) [198]. To detect exogenous apocynin-induced cellular ROS, in vitro incubation of apocynin and DCFH is required (to prevent fluorescence interference owing to reaction between them) [199]. Extracellular hydrolysis of DCFHDA reduces the difference in esterase activity among cells, which assists in improved detection of ROS level. Instead of direct reaction with H_2_O_2_ or superoxide, the DCFH reacts with H_2_O_2_ or cytochrome c in presence of transition metals [200]. Although sensitivity of DCFH to ONOO is higher in comparison to oxygen free radicals, DCFDA is also sensitive to oxygen radicals with high detection range of ROS in comparison to DCFHDA. The DCFDA is also reported as cell permeable, where enzymatic cleavage of diacetate (DA) by esterase is like DCFHDA cleavage. As oxidation into DCF can be through O_2_˙^−^, H_2_O_2_, or NO, this reveals that DCFDA is reactive against both nitric oxides and ROS [201]. The DCFHDA is apt to react with peroxynitrone or H_2_O_2_, whereas DCFDA recognizes broad range of radicals. Therefore, care must be taken while interpreting DCFDA data. DCFHDA offers several problems, such as while detection of H_2_O_2_, the DCF fluorescence intensity is indirectly related with specific free radical. The radicals such as NO_2_^−^, ONOOH, ONOO^−^, and HOCl may oxidize DCFH into DCF, thereby interfering in H_2_O_2_ level determination [202]. In addition, the DCF˙^−^ is reported to react with oxygen to form additional O_2_˙^−^ and H_2_O_2_, which further adds to H_2_O_2_ results [203]. Even in absence of H_2_O_2_, the oxidation of DCFH may be stimulated by various factors (such as: cytochrome c, peroxidases, horseradish peroxidase (HRP), ferrous ion, and hematin) [204]. The indiscriminating recognition ability of DCFHDA for various oxygen radicals and oxidants, makes it non-competent for specific detection in cells. After cautious fixing of control groups and rational result analysis, it is possible with DCFHDA to detect changes in cellular OS by extra or intracellular stimulation.

#### 7.1.3. Amplex Red (AR)

Amplex red (AR) is a sensitive probe that is used specifically to measure extracellular H_2_O_2_ with detection limit of approximately 5 pmol. AR reacts with H_2_O_2_ via HRP catalyzed one-electron pathways to produce fluorescent resorufin (with Ex and Em of 563 and 587 nm, respectively), whose intensity reflects concentration of H_2_O_2_. Resorufin is a stable product which allows detection of H_2_O_2_ under oxidative and reductive conditions. It presents a broad range of applications, for example: in detection of mitochondrial-generated H_2_O_2_ during mitochondrial gene damage and ATP synthesis in presence of Lon (La) protease [205]. AR selectivity for H_2_O_2_ is challenged by various factors. For example, in presence of peroxynitrite or peroxynitrite-induced radicals, the AR is catalyzed by HRP that produces resorufin at a faster rate in comparison to H_2_O_2_, which necessitates specific inhibitors [206]. Additionally, the HRP may react with electron donor NADPH and GSH (reduced form) to yield H_2_O_2_, that results in further production of resorufin. Moreover, the resorufin may be reduced by NADPH-CYP450 reductase into a non-fluorescent and colorless compound, along with generation of O_2_˙^−^or H_2_O_2_, that creates a problem in the identification of H_2_O_2_ [207]. The low affinity of AR towards cell membranes creates a problem in the precise measurement of intracellular H_2_O_2_ [208]. Moreover, AR can be converted into resorufin in various tissue samples (kidney and liver) by carboxylesterase without prerequisite for HRP, oxygen, or H_2_O_2_; that creates the need for re-investigation for comparable assays [209]. Based on radicals and enzyme interference, the AR is considered a challenging probe for the measurement of intracellular H_2_O_2_. However, AR can be applied in cell-free systems in presence of HRP wherein H_2_O_2_ is released from cell or split mitochondrial particles.

### 7.2. Fluorescent Protein-Based Redox (FPBR) Analysis

Fluorescent protein-based redox (FPBR) probes are formulated by combining fluorescent and prokaryote redox sensitive proteins. The recombinant proteins are administered to the cells via adenovirus or plasmid and targeted on subcellular organelles, to determine the redox state of certain regions [210]. Recomb-proteins redox dependent fluorescence change is attained by change in structure of disulfide bonds and main chain in oxidized conditions. The FPBR probe affords real time and dynamic detection of change in redox potential of reaction (that involve radicals) without special need for permeation in target cells. The fluorescent proteins (FP) reside in cells and allow long-term detection of cellular radicals. The combination of FP with targeting signal peptide or retention sequence allows FP to target various organelles, and therefore reveals redox status. For redox detection, the FPRB analysis involves numerous colors of redox-sensitive targeting proteins.

#### 7.2.1. Redox-Sensitive Green Fluorescent Protein (roGFP)

Formulation of redox-sensitive green fluorescent protein (roGFP) involves addition of redox reactive cysteine in GFP beta-strand 7 and 10 at site Q204 and S147. Under a reduced environment, it causes formation of disulfide linkage between two domains, which reacts to redox changes in intra/extra-cellular systems, leads to reversible ratiometric change in the intensity of fluorescence [211]. Based on roGFP, numerous probes are developed.

##### roGFP1 and roGFP2

roGFP 1 and 2 are the first two analogues of roGFP that are differentiated based on the amino acid (T65S) unit. These may indicate conversion of dithiol/disulfide that is stimulated through ROS accumulation. These are used to examine variation in thiol/disulfide equilibrium [149]. The cysteine pair of roGFP 1 and 2 is protonated under physiological pH. Rather than direct ROS measurement, roGFP1 and 2 determine dynamic redox change. As roGFP1 and 2 undergo complete oxidation by oxidating organelles (lysosomes and endosomes), they can be appropriate for reduced environments (mitochondria, cytoplasm, peroxisomes, and nucleus) [212]. The sensitivity of roGFP also depends upon pH and speed of reaction. As intensity of roGFP fluorescence does not change quickly with change in redox condition, this indicates suitability of roGFP in monitoring the long-term redox shifts [210]. Moreover, in comparison to roGFP1, roGFP2 is easily influenced by variation in pH (ranged between 6 to 8), hence detection condition is an important aspect to consider.

##### roGFP1-iX

Depending upon roGFP1 usage in oxidizing environment, the roGFP1-iX was formulated. The formulation of roGFP1-iX involves incorporation of the amino acids next to cys147 and mutation of H148S in roGFP1 beta-strand 7. The roGFP1-iX offers fast reaction speed and low pH sensitivity (ranged from 6 to 8) in comparison to roGFP1. This results in roGFP1-iX suitability in monitoring redox in oxidating organelles such as endoplasmic reticulum (ER) [213,214].

##### roGFP1-iL and roGFP1-RX

roGFP1 probe was modified to roGFP1-iL for sensing the redox of ER. The roGFP1-iL disulfide bonds are partly oxidized, which generates high reduction potential to examine the variation in redox condition [215]. roGFP1-iL and Grx1 in combination exhibit high sensitivity towards 2GSH or GSSG. In comparison to roGFP1, roGFP1-RX exhibits high reaction speed and dynamic range through addition of three amino acids (carrying positive charge) next to cysteine [216].

##### Grx1-roGFP2-iL and roGFP2-Orp1

Just like roGFP1, derivatives were also created for roGFP2. The broad midpoint potential of Grx1-roGFP2-iL makes it suitable for determination of redox in ER and cytosol. In comparison to roGFP2, the derivative Grx1-roGFP2-iL exhibits higher specificity in measurement of redox potential of GSH [192]. The derivative probe roGFP2-Orp1 is H_2_O_2_-specific. roGFP2-Orp1 was created by combining roGFP2 with yeast peroxidase Orp1, and redox relay equivalent between the two allows the roGFP2-Orp1 probe to efficiently indicate the H_2_O_2_ level [217]. The pH stability of this derivative probe makes it applicable for sensing alteration in H_2_O_2_ levels in mitochondria and cytosol in micromole [218]. The roGFP2-Orp1 probe exhibits superior selectivity towards H_2_O_2_. It is important to note that none of the studies reported roGFP probes-based analysis to determine the concentration or net formation of H_2_O_2._ In fact, roGFP probes can only sense the variation in H_2_O_2_ or redox level instigated by external influences.

#### 7.2.2. Redox-Sensitive Yellow Fluorescent Protein (rxYFP)

These are created using yellow shifted GFP-derived protein (YFP), wherein the cysteine pair is incorporated next to chromophoric domain, thereby forming a reversible disulfide bond. The oxidative environment leads to a change in spatial conformation of YFP that causes decrease in fluorescence intensity (near 527 nm) which allows visualization of cellular redox in vivo [219]. Attributed to suitable midpoint redox potential, the rxYFP can be used to sense the dynamic change in GSH, GSSG, thiol, and disulfide in various reducible cellular regions (such as cytosol, mitochondria, and nucleus) [220]. Generally, rxYFP targets to examine the whole redox change rather than sensing of particular redox couple.

##### rxYFP-Grx1P

This is the modified probe of rxYFP, which is engineered by combining glutaredoxin-1 of yeast (Grx1p) and rxYFP. This probe is suitable for sensing the redox potential of intracellular GSH [221]. This is more specific to GSH in comparison to H_2_O_2_, hydroxyethyl disulfide, and cysteine.

##### rxYFP 3R

This protein includes addition of three more positive charged cysteine residues; that is, 200R/204R/227R in rxYFP results in formation of rxYFP3R, that exhibits 13 times higher reactivity for GSH when compared with rxYFP [222]. It is important to note that rxYFP modified probes exhibit enhanced specificity for 2GSH and GSSG, and improved stability at physiological pH. rxYFP exhibits pH sensitivity and their chromophores pKa changes as per the pH of environment. In neutral environments, quenching of fluorescence occurs on the rxYFP protonated form, which the reduces fluorescence signal by 2.2 times with nonsignificant excitation wavelength shift [223]. However, this probe is unsuitable for ratiometric quantification of redox.

#### 7.2.3. HyPer

This is H_2_O_2_ sensing cpYFP-OxyR recombinant protein. HyPer is engineered by incorporation of circularly permuted yellow fluorescent protein (cpYFP) into regulating domain (RD) of prokaryote H_2_O_2_ sensing OxyR protein. This creates disulfide linkage between reactive Cys208 and Cys199 of RD on reaction with H_2_O_2_; which alters conformation of OxyR, that further changes fluorescence intensity and conformation of whole protein and thus correlates alteration in level of H_2_O_2_ with intensity of fluorescence. HyPer is applicable to various cellular systems and living organisms (caenorhabditis, arabidopsis, yeast, elegans, and mouse) to study intracellular dynamic change and real time analysis of H_2_O_2_ [224,225]. In comparison to roGFP2-Orp1, the HyPer exhibits high reaction speed and sensitivity towards H_2_O_2_, attributed to position of redox sensitive cysteine pair in HyPer [226]. To effectively detect the change in H_2_O_2_ level, the oxidized HyPer must reduce in time, which is mediated through intracellular GSH [227]. Therefore, HyPer can be used to determine equilibrium between GSH and H_2_O_2_.

##### HyPer-2 and HyPer-3

These are the HyPer derivative probes that are created via mutation of A406V (for HyPer-2) followed by mutation of H34Y (for HyPer-3) in HyPer [228,229]. HyPer-2 exhibits higher stability than HyPer-3 amongst monomers in OxyR dimers, whereas HyPer-3 exhibits faster response time and reaction speed towards H_2_O_2_. Both HyPer-2 and 3 exhibits broad dynamic range as compare to HyPer; and used for fluorescence lifetime in vivo imaging of H_2_O_2_. The change of pH environment is a major challenge for HyPer probes. As in cells the H_2_O_2_ concentration depends upon disproportion of O_2_˙^−^, and an increase in pH may decrease dismutation rate of O_2_˙^−^, while using such sensors one must focus systemic or local pH.

#### 7.2.4. Circularly Permuted Green Fluorescent Protein (cpYFP)

The probe of cpYFP (formerly pericam) was initially used to detect calcium, but exertion of ratiometric fluorescence flash by cpYFP/Pericam due to rise in O_2_˙^−^ level proved cpYFP as an effective tool for imaging of O_2_˙^−^ [230]. Binding of cpYFP to mitochondrium targeting sequence, permits cpYFP to sense subcellular level ROS [231]. One study [232] claims cpYFP responses to variation in pH, which suggests cpYFP specificity towards O_2_˙^−^ as point for further discussion. It is important to notice that as cpYFP fluorescence intensity is influenced by mitochondrial matrix alkalinity, hence precaution must be taken while imaging mito-O_2_˙^−^.

#### 7.2.5. HyPerRed (rxRFP)

This gene encoded red fluorescent probe assists in detection of H_2_O_2_. This HyPerRed probe is engineered by substitution of sensing domain of calcium probe R-GECO1 with OxyR. The disulfide bond formation between Cys208 and Cys199 develops HyPerRed specificity for H_2_O_2_ identification [233]. The sensitivity and kinetics of HyPerRed for H_2_O_2_ is just like HyPer probe. HyPerRed has the ability to detect H_2_O_2_ in low concentration, and its high sensitivity to change in pH necessitates for adjacent analysis, such as HyPer-C199S or HyPerRed-C199S, which are used standard probe for variation in pH [233]. Apart from that, the non-ratiometric reaction among H_2_O_2_ and HyPerRed (which limits HyPerRed application for quantification) necessitates other GFP sensors as controls for quantification assays.

##### TrxRFP1 and cpRFP

Based on redox-RFP, several other fluorescent biosensing probes have been constructed. In order to monitor the redox dynamics in case of thioredoxin and thiol/disulfide transformation, two probes, namely TrxRFP1 and cpRFP, were engineered; they assist in the analysis of variation in local or general redox-state in mammalian cells [234,235]. The constructed RFP probes assist in analysis of various redox states and free radicals, but RFP probes also offer some limitations. As construction of RFP probes is a complicated process, during analysis the potential factors must be monitored. For example, the nature of target recipient cells may affect the RFP expression level that in turn impacts the probe capability. The roGFP undergoes reversible oxidation on exposure to oxygen and is less sensitive to oxidants in vivo in comparison to in vitro, attributed to AO presence [236]. The rate of reaction between these RFP proteins and redox substances are slow, which causes less immediacy in intra-cellular radical determination. Just like roGFP2-Orp1 and HyPer, the selectivity of these RFP probes against H_2_O_2_ is based on a reversible oxidation reduction that is facilitated by H_2_O_2_ and 2GSH/GSSG couple. However, their reaction capacity difference may cause inaccurate data analysis. As protein-based biosensors offer difficulty in quantitative analysis of oxygen free radicals, it necessitates calibration and verification during analysis. The fluorescence analytical tool also imposes high impact over protein-based biosensors data. Use of microscopy with fluorescence can select various types of cells, but it is unsuitable for large quantity sample imaging. Flow cytometer is commonly used in mammalian cell analysis, whereas laser scanning plate readers are applicable for tissues with high throughput screening (HTS) [237]. Although fluorescent microplate readers offer fluorescence data, they cannot generate information at the cellular or sub-cellular level.

### 7.3. Chemiluminescence Analysis (CLA)

Same as fluorescence analysis, chemiluminescence analysis (CLA) probes are commonly used to detect the O_2_˙^−^. These are sensitive to radicals and offers ease of handling. The CLA probes react with O_2_˙^−^ to produce photons that are captured by photometer without need for exciting light source [238].

#### 7.3.1. Lucigenin

Attributed to its good membrane permeation property, lucigenin or LC^2+^ or bis-N-methyl acridinium nitrate is reported to measure O_2_˙^−^ (generated by macrophages, neutrophils and extracellular isolated enzymes) [239]. A study validated LC^2+^-assisted O_2_˙^−^ detection in various cellular and enzymatic system (such as lipoamide dehydrogenase, isolated/intracellular mitochondria, xanthine oxidase, and phagocytic NADPH oxidase) [240]. LC^2+^ is reported to offer several limitations. This is based on the fact that flavoprotein reductase reduces LC^2+^ into LC˙⁺, that further reacts with O_2_ to form O_2_˙^−^. Apart from O_2_˙^−^, other reducing moieties or nucleophiles (such as alkaline H_2_O_2_) may also cause luminescence of LC^2+^ via generation of radical LC˙⁺ [241]. The reductase-mediated formation of LC˙⁺ and lucigenin is sensitive to change in reductase activity. The low reaction between LC^2+^ and O_2_˙^−^ suggests LC˙⁺ inability to detect O_2_˙^−^ radicals at lower level.

#### 7.3.2. Luminol (LH)

Although 5-amino-2,3-dihydroxy-1,4-phthalazinedione or luminol is extensively used for detection of O_2_˙^−^, it responds to several types of free radicals such as O_2_˙^−^, ˙OH, H_2_O_2_, and ONOO^−^ [242]. The luminol (LH^−^) gets oxidized into luminol radical (LH˙). This nonspecific conversion not only includes O_2_˙^−^ but also involves more competent oxidants such as ˙OH and CO_3_˙^−^. In comparison to O_2_˙^−^, the LH˙ possesses lower rate of reaction with O_2_ to generate O_2_˙^−^, which hampers O_2_˙^−^ detection [243]. Additionally, the AO inside cells may directly react with LH˙ or may compete with radicals which further reacts with LH˙. Hence, luminol offers the benefit of testing AO capacity. Generally, at neutral pH the luminol exhibits inefficient reaction with free radicals. The wavelength of emitting light of luminol and its derivatives is 400 nm, which allows auto-absorption of biomolecules in cells or biological systems. The nonspecific recognition of different free radicals by luminol limits its applications. The CL of luminol involves complex and uncontrolled factors, so it is difficult to examine the free radical generation in cells or biosystem alone with this probe. However, by luminol presence of O_2_˙^−^ can be determined without preventing other free radicals.

#### 7.3.3. Luminol Analogue (L012)

The 8-amino-5-chloro-2,3-dihydro-7-phenylpyrido [3,4-d] pyridazine sodium salt or luminol analogue or L012 assists in analysis of O_2_˙^−^ in various body cells and biological systems. The L012 exhibits high sensitivity and detection capacity for O_2_˙^−^ [244]. The L012 does not react directly with O_2_˙^−^. It generates a non-specific intermediate LH˙ in its first conversion. This LH˙ may react with O_2_ to generate O_2_˙^−^ and L012 quinone. In presence of H_2_O_2_, the L012 quinone forms LH˙–OOH that leads to false results of O_2_˙^−^ [181]. In comparison to other methods, the CL offers high reaction speed towards O_2_˙^−^, that offers an advantage of detection at low O_2_˙^−^ level. Hence in cells or systems, the CL offers sensitive monitoring against generated O_2_˙^−^ [245].

### 7.4. Electro-Chemical Biosensing (ECB)

Formulation of electro-chemical biosensors involves application of alternate polyaniline-sulfonic acid and cytochrome-c layers over gold wire electrode for stable, sensitive, and selective quantification of O_2_˙^−^. Principally in ECB, the O_2_˙^−^ can decrease some proteins and at suitable potential, such proteins can be re-oxidized using an electrode. Such process generates electric current signal that is captured using sensor; and based on the proportion of O_2_˙^−^ and electric signal strength, the concentration of O_2_˙^−^can be determined [246]. Electro-chemical biosensing (ECB) is used to analyze the real-time formation of O_2_˙^−^, and to detect in vitro O_2_˙^−^reaction with specific AO [247]. The biosensing electrode quantifies O_2_˙^−^ in vitro and in vivo with higher efficiency, and combination of electrode with cytochrome-c enhances the selectivity of cytochrome-c. ECB suffers with the availability of limited proteins for coating over the electrode surface. One study [248] suggests immobilization of proteins and cytochrome-c in layer-by-layer pattern over electrodes exhibits high sensitivity in comparison to mono-layer.

### 7.5. Chromatographic Analysis

Chromatography is one of the most widely coupled technique used for the analysis of OS. Liquid chromatography (LC), gas chromatography (GC), ultra-performance liquid chromatography (UPLC), and high-performance liquid chromatography (HPLC) are the commonly used techniques for OS determination. Chromatography can be coupled with various mass spectrometry (MS) for effective determination of OS. Some important mass coupled techniques of chromatography to measure OS include UPLC-MS, GC-MS/MS, LC-TOFMS (time of flight mass spectrometry), and HPLC-TOFMS. However, GC-MS, HPLC-MS, LC-MS, and LC-MS/MS are most applied techniques for the quantification of ROS [249]. Chromatographic analysis is applied to separate and identify the ˙OH radicals and related products. Principally, the ˙OH radicals react with particular reagent to form stable compounds that can be detected with chromatographic assay (mostly liquid chromatography together with mass spectrometer). Some important reagents for free radical stabilization include DMSO, salicylic acid, and benzoic acid. The salicylic acid when reacting with ˙OH forms dihydroxy-benzoic acid (DHBA) that is quantified by high performance liquid chromatography (HPLC) using electrochemical detector. This supports chromatography ability for in vivo analysis of ˙OH [250]. The HPLC can be used in detection of ˙OH or AO activity in several tissues and reaction system [251]. Although chromatographic analysis is very sensitive, fast, and efficient in detecting the ˙OH, the too complex pre-detection treatment, process of reaction, and reaction products altogether limit the wider use of chromatography. Based on non-specificity of fluorescent and luminescent probes for ˙OH, chromatography can be applied to detect ˙OH presence.

### 7.6. Spectro-Photometric Analysis

For ROS detection, spectrometric analysis (the time-based technique) works over reaction among redox materials and radicals and measures the absorbance difference at distinct wavelengths among substrates and products; which thereby assists the semi-quantification of free radicals. The spectrophotometric analysis includes various assay methods for detection of ROS given as follows:

#### 7.6.1. Cytochrome c Reduction Assay

Assay of cytochrome c reduction (CCR) assists in analysis of O_2_˙^−^. Cytochrome c absorbance is detected at 550 nm. One study [252] suggests a relationship between production of O_2_˙^−^ and alteration in physiological functions in perfused rat. Reduction of cytochrome c is attributed to exhaustion which suggests relation between tissues activity and O_2_˙^−^. Specificity of CCR towards different radicals is relatively less. One study suggests that enzymes/reductants such as GSH and ascorbate can reduce cytochrome c, and therefore interfere in O_2_˙^−^ identification [204]. Presence of H_2_O_2_ and peroxynitrite (ONOO) interferes in O_2_˙^−^ detection, attributed to their ability to reoxidize cytochrome c (reduced) into its original form. However, administration of inhibitors or scavengers (urate for ONOO or catalase for H_2_O_2_) can prevent such re-oxidation [253]. The exogenous O_2_˙^−^ supports in determination of specificity of CCR towards O_2_˙^−^, and O_2_˙^−^quantification can be indirectly analyzed from extent of inhibition of CCR by O_2_˙^−^. For example, production of O_2_˙^−^ from NADPH oxide can be detected through O_2_˙^−^ inhibition rate of CCR [254]. One study suggests that cytochrome c succinylation or acetylation improves the CCR specificity to O_2_˙^−^, whereas at same time the rate constant may decrease [255]. The large molecule size and charge strength of cytochrome c affect its intracellular sensing ability that limits sensing of low concentration O_2_˙^−^ produced by mitochondria and cytoplasm in the cell [204]. As in comparison to other probes, CCR exhibits lesser sensitivity and selectivity towards O_2_˙^−^ detection; hence, it necessitates the use of specific scavenger/inhibitor for elimination of radicals.

#### 7.6.2. Nitro Blue Tetrazolium (NBT) Assay

Nitro blue tetrazolium (NBT) is another often employed probe to detect O_2_˙^−^. This may undergo reduction with peroxidase or dehydrogenase and reaction with O_2_˙^−^ to form diformazan (that exhibits absorbance at 620 nm corresponding to concentration of O_2_˙^−^). The NBT detects O_2_˙^−^ generated from monocytes and macrophages [256]. The specificity and sensitivity are major challenges to new dyes. As per NBT susceptibility to intracellular reductase, the NBT may not be specific to single radical detection.

#### 7.6.3. Aconitase Inactivation Assay

The catalyzing activity of aconitase (which exist in mitochondria and cytoplasm) towards citrate and isocitrate conversion is inactivated by O_2_˙^−^. This occurs via reversible loss of Fe from its cubane (4Fe–4S) group, that leads to change in aconitase absorbance at 240 and 340 nm [257]. The regular O_2_˙^−^ generation may lead to conversion of aconitase between active and inactive form. This process is affected by rate of O_2_˙^−^ formation, that allows determination of variation in concentration of O_2_˙^−^ [258]. One study suggests that coupled spectrophotometric detection of cisaconitate and isocitrate at 240 and 340 nm assists in measurement of O_2_˙^−^ generation in presence of NADPH [259]. The fast speed and sensitivity of reaction of aconitase with O_2_˙^−^ allows wide use of this assay in various cells (fibroblasts and macrophages) and tissues (heart, brain and liver) [260]. However, the oxidants such as NO, O_2_, and H_2_O_2_ may also inactivate aconitase (but at very low rate as compare to O_2_˙^−^), thereby interfering in specific O_2_˙^−^ detection. Therefore, while detecting O_2_˙^−^ using aconitase method, one must eliminate the interference of other oxidants with specific inhibitors [261].

#### 7.6.4. Boronates Assay

Synthetic boronates may react with ONOO radical and H_2_O_2_, thereby rapidly generating a single stable phenolic product [262]. These are very effective to monitor and quantify the ONOO and H_2_O_2_ in cells. These may offer more sensitive and selective detection by attaching with a fluorophore (that forms fluorescent product upon reaction with H_2_O_2_) [263]. Facts suggest synthesis of highly selective boronates-based fluorescent probes for H_2_O_2_ fluorescence imaging in cells and tissues; for example: peroxyfluor-1 (PF1), peroxy crimson 1 (PC1), and peroxy green 1 (PG1) [264]. Engineering of combined probes makes the boronate assay more suitable to detect and monitor the H_2_O_2_. For example, mitochondria peroxy yellow 1 (MitoPY1) comprising chemo selective boronate-based switch and mitochondrial-targeting phosphonium group undergoes reaction with H_2_O_2_ to generate highly fluorescent MitoPY1ox that signals its fluorescence at 528 nm [265].

#### 7.6.5. Diaminobenzidine (DAB) Assay

The colorimetric indicator diaminobenzidine (DAB) (engineered for ROS imaging) when it reacts with H_2_O_2_ under peroxidase catalysis forms a brown precipitate (insoluble), that can be imaged microscopically in cells or tissues [266]. The DAB probe offers the benefit of polymeric products’ stability during variation in temperature and illumination [267]. Attributed to stability and insolubility of DAB product, the DAB can be detected precisely in cells, which supports ROS detection. As DAB method is economical, easy to operate, and demands no strict requirements for lab working conditions, it is proposed as one of the best methods to detect ROS at subcellular level.

### 7.7. Electron Paramagnetic Resonance (EPR) or Electron Spin Resonance Assay

The electron spin resonance (ESR) or electron paramagnetic resonance (EPR) assay allows direct detection of oxygen free radicals. The short life of oxygen free radical necessitates specific spin traps to furnish stability (via inclusion of radicals into their structures or oxidation to form stable radical) and ease of capturing. The capturing agents are the important elements of ESR or EPR assays. For free radical analysis among various spin traps, the 5,5-dimethyl-1-pyrroline-N-oxide (DMPO) is considered the most important probe. Principally, in this assay on reaction with ˙OH and O_2_˙^−^, the DMPO form adducts DMPO/˙OOH and DMPO/˙OH correspondingly; which are further detected by ESR spectrometer to produce their spectra. The DMPO-based ESR/EPR method can analyze the radicals in both cell and cell free system. A study highlighted that DMPO/˙OOH can be sensed in NaOH/H_2_O_2_/Fe(III) system; which exhibits production of O_2_˙^−^ and exposes the Fe(III) need for generation of O_2_˙^−^ [238].

There are some important aspects to be considered about DMPO-based detection in the presence of ROS. For example, in presence of a reducing agent (such as ascorbate), the O_2_˙^−^ and DMPO adducts may undergo back reduction, which generates false data [238]. Another aspect is that in an aqueous system, the DMPO ability for ˙OH and O_2_˙^−^ detection is impaired [268]. Another area of concern is that in cells or tissues detection of DMPO, adducts can be obstructed by DMPO/˙OOH short life, transition metals presence, DMPO/˙OOH degeneration to DMPO/˙OH, and further reaction with DMPO/˙OH or DMPO/˙OOH [269]. Hence, during interpretation of data (using DMPO spin trap) all these mentioned factors must be considered.

The 5-(Diethoxyphosphoryl)-5-methyl-1-pyrroline N-oxide (DEPMPO) is one of the DMPO analogues that forms a more stable adduct with O_2_˙^−^, due to which secondary radical adduct data also affect ESR spectral interpretation [270].

5,5-dipropyl-1-pyrroline 1-oxide (DPPO) and 5-butyl-5-methyl-1-pyrroline 1-oxide (BMPO) are the two cyclic nitrone spin traps. The more lipophilic and stable radical adducts of DPPO and BMPO makes them widely applicable [271]. Unlike DMPO/˙OOH, the adduct of BMPO-O_2_˙^−^ does not proceed non-enzymatic conversion into BMPO hydroxyl adduct, thereby making detection easy and credible in comparison to DMPO. Moreover, spectra of BMPO-radical adduct offers high signal to noise ratio during detection of O_2_˙^−^, ˙OH, and other radicals, in comparison to DMPO [272].

4-phosphonooxy-2,2,6,6-tetramethylpiperidine-N-hydroxyl (PP-H), 1-hydroxy-3-carboxy-pyrrolidine, 1-hydroxy-2,2,6,6-tetramethylpiperidin-4-yl-trimethylammonium (CAT1-H), and 1-hydroxy-2,2,6,6-tetramethyl-4-oxo-piperidine (CP-H) are three cyclic hydroxyl-amines that are widely used in ESR assay [273]. PP-H, CAT1-H, and CP-H offers firm spin trapping as they can be oxidized via free radicals and generate stable radicals with a half-life of several hours that allows detection by EPR [238]. Different charged probes exhibit distinctive membrane permeabilities and lipophilicities; so combined use of probes benefits in using ESR analysis to detect various sources of cell radicals [189]. Extracellular or intracellular generated O_2_˙^−^ can be differentiated simultaneously using a set of mentioned probes. Among all probes, CAT1-H and PP-H can only sense O_2_˙^−^ in mitochondria, and TEMPO-H assists in detection of O_2_˙^−^ in separate mitochondria particles; whereas CM-H (1-hydroxy-3-methoxycarbonyl2,2,5,5-tetramethylpyrrolidine) and CP-H assists in O_2_˙^−^detection in cytoplasm [274]. There are several factors that affect ESR detection, among which pH affects radical generation and mutual transformation; whereas temperature affects radical energy level; while a liquid environment is responsible for radicals’ resonant state [275]. Hence, to reduce non-experimental interference, care must be taken for the mentioned conditions. The ESR method can be applied for identification of oxygen, nitrogen, or organic free radicals of non-cellular and even non-biological systems.

Any aforesaid method has ability to detect the ROS level, but for credible results the fair utility of each method based on experimental requirements is recommended. For example, the probes for fluorescence and luminescence can detect the majority of oxygen radicals qualitatively and semi-quantitatively. High sensitivity, flexibility, and simplicity of these probes enhances their application in various cells [276]. These probes can detect short life free radicals in cells. ROS generation and action in cells is extremely regional and attributed to their low level, and overlapping reactivity creates difficulty in measurement. Apart from DAB, classical fluorescent probes are developed for improved detection of localized free radicals. Depending upon MitoTracker and DHE combination, several probes are used for detection of localized mito-ROS [277]. For example, nuclear peroxy emerald 1 (ucP1) was engineered and developed to detect nuclear H_2_O_2_, and for enrichment of fluorescent probes in organelles. Genetically coded proteins are used to monitor organelle-generated ROS and targeted on selective organelles by linking specific signal peptide and retention sequence. For example, recombinant HyPer proteins are cloned in different cell organelles reacts with localized H_2_O_2_ with different reactivities, which determines H_2_O_2_ generation and removal. The different redox states and pH environments in each organelle are important factors affecting the activity of genetically encoded fluorescent probes [278]. Probes of fluorescent proteins can detect redox changes at subcellular level. The DHE and cytochrome c methods are simultaneously applied for detection of NADPH oxidase (NOX) generated O_2_˙^−^ in the presence of palmitate. Among these cytochrome c assists in qualitative detection, whereas DHE assist in quantitative detection [279]. The combined DCF and confocal assay can simply detect total ROS, whereas ESR can determine single radical presence. The analytical methods for ROS detection are developing very rapidly and applied successfully, however efforts are needed to minimize the weaknesses associated with such methods. Table 1 presents various probes, target ROS, as well as pros and cons of different ROS detection methods.

## 8. Conclusions

The understanding of dual beneficial and harmful effects of ROS and antioxidants on the tumor processes would assist in resolving many contradicting impacts of such moieties in different investigations in the future. Current “state-of-the-art” endorses the notion that there exist various ROS pools that differ in actions. For example, on one hand the ROS derived from NADPH oxidase promotes proliferation in the mouse intestinal cells, whereas on the other hand ROS generated from loss of TIGAR causes impairment of proliferation in the same cells. It becomes imperative to understand that extracellularly NADPH oxidase produces O_2_˙^−^, however intracellularly TIGAR supports PPP and thereby safeguards from ROS. Moreover, while translating the AO and whole-body gene knockout studies, it becomes essential to consider the role of AO systems in micro-environments. The population of cells may depend upon both AO programs and ROS generation for functioning. Furthermore, it is not necessarily the case that ROS harmful or beneficial effects in cells will be mutually exclusive. Present studies recommend that improving the technology (such as genetic screening) which brings higher clarity in determination of enzymatic pathways and scaleup in cancer models profiling using omics technology would definitely develop the in-depth understanding of AO pathways and ROS complexities.

This review highlights various methods to measure the concentration of ROS, such as spectrometry, spectrophotometry, chemi-luminescent and fluorescent probes, chromatography, and genetic encoding based fluorescent protein assay. The current review compared and emphasized various merits and demerits of each ROS detection method. As per the findings, it was revealed that any aforesaid method has ability to detect the ROS level, but for credible results the fair utility of each method based on experimental requirements is recommended. Combination of multiple analytical methods is best approach based on following facts: (i) achievement of complete assistance of qualitative and quantitative analysis, (ii) single radical identification among wide range of ROS, (iii) for ROS screening simple methods can be applied, and for additional confirmation complex assay can be used. In spite of several attempts in development of detection methods for free radicals, the precise measurement of ROS in cells and tissues is a big challenge attributed to very short life and low concentration of ROS. To select the suitable means for measurement of a specific radical, one must consider system characteristics, ROS production level, free radical properties, and transformation among them. The present review recommends that although numerous methods for ROS detection are developing very rapidly and applied successfully, still more efforts are required to minimize the limitations associated with currently available methods.

## Figures and Tables

**Figure 1 antioxidants-10-00128-f001:**
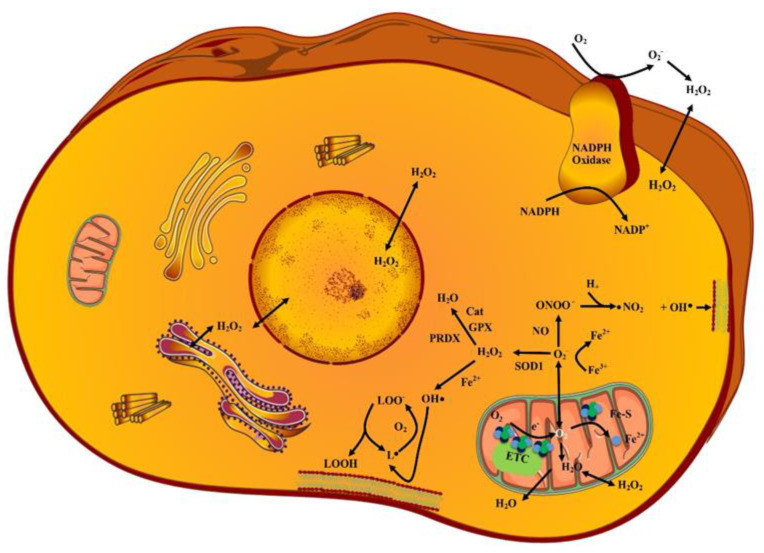
Various types of reactive oxygen species (ROS) and their sources.

**Figure 2 antioxidants-10-00128-f002:**
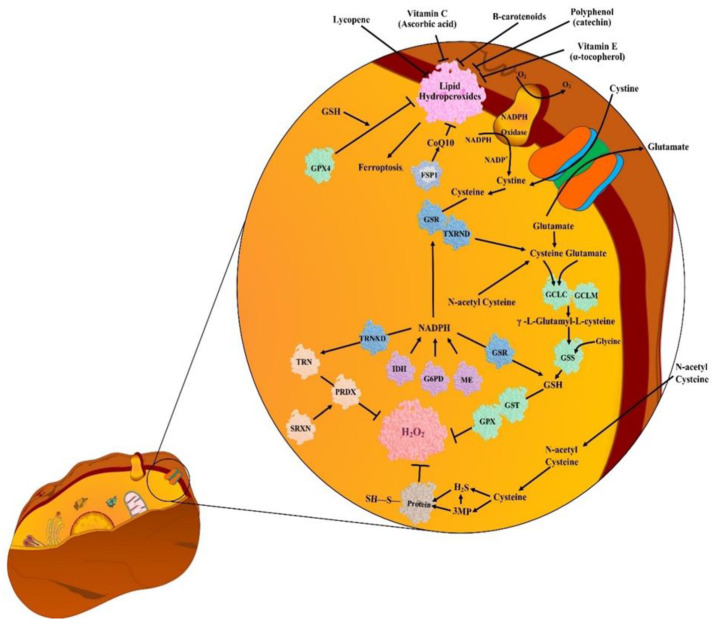
Various types of endogenous (EnAO) and exogenous (ExAO) antioxidants.

**Figure 3 antioxidants-10-00128-f003:**
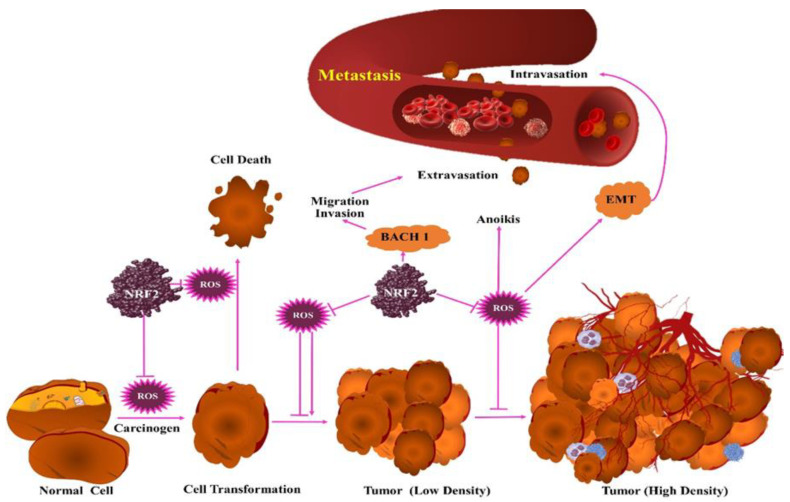
Role of NRF2 at various of cancer stages.

**Table 1 antioxidants-10-00128-t001:** The probes, target ROS, pros, and cons of ROS detection methods.

Detection Methods	Probes	Target ROS	Pros	Cons
Fluorescent Chemicals	Dihydroethidium Dichlorodihydro fluorescein Amplex Red	O_2_˙^−^ and H_2_O_2_	Cell permeation possible Stable products are formed Intensity can be quantified	Formation of complex product Specificity is low Interference occurs by OONO^−^
Fluorescent Proteins	Green fluorescent proteins (roGFP1, 2, iL, P1-RX), Grx1-roGFP2-iL and roGFP2-Orp1 Yellow fluorescent proteins (rxYFP-Grx1P, rxYFP 3R) HyPer (HyPer-2 and HyPer-3) Circularly permuted green fluorescent protein (cpYFP) HyPerRed (rxRFP), TrxRFP1 and cpRFP	H_2_O_2_ and Variation in redox level	Offers real-time and dynamic detection of change in redox level with no special requirement for cell permeability As fluorescent proteins can stay in cell, so allow long term detection of cell radicals Reveals localized redox status Cell friendly	Preparation of these probes is complicated roGFP exhibits low sensitivity Reaction rate of proteins with redox substance is quite low Differential selectivity of roGFP2-Orp1 and HyPer for H_2_O_2_ due to different reaction capability may lead to data inaccuracy
Chemi-luminescence	Lucigenin Luminol Luminol analogue	H_2_O_2_ and O_2_˙^−^	Cell permeation possible	Sensitivity low Selectivity low Unstable intermediates
Electro-chemical Biosensing	Alternate polyaniline-sulfonic acid and Cytochrome-c layers over gold wire electrode	O_2_˙^−^	Detection is fast Sensitivity is high	Preparation is complex
Chromatography	DMSO Salicylic acid and Benzoic acid	˙OH	Detection is fast Sensitivity is high	Complex products are formed
Spectro-photometry	Cytochrome c Nitroblue tetrazolium Aconitase Boronates Diaminobenzidine	O_2_˙^−^ and H_2_O_2_	Sensitivity is high Detection is fast Single product is formed	Specificity is low
EPR/ESR	Spin traps	ROS and RNS	Sensitivity is high Specificity is high	More expensive

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
