# Peer review of "Comprehensive Review of Methodology to Detect Reactive Oxygen Species (ROS) in Mammalian Species and Establish Its Relationship with Antioxidants and Cancer"

_antioxidants, 2021, doi:10.3390/antiox10010128_

Round 1

Reviewer 1 Report

This review provides considerable valuable intellectual perspective on a complex topic. There are many references on antioxidant functions and responses in mammalian tissues. While there may be some cross-application, review of plant systems are essentially non-existent. One suggestion may be to indicate this review is primarily for mammalian free radicals and antioxidants. For example, change the title:

Comprehensive review of methodology to detect reactive oxygen species (ROS) in mammalian species and establish its relationship with antioxidants and cancer

The graphics are very good and useful to assist following the review comments. In some instances clarity may be improved by reducing wordiness of the text.

Some observations and comments suggest improvements in: adherence to agreement between singular/plural structure; choice of ‘article’ or no use of article; elimination and or choice of unnecessary modifying phrases, avoid use of “Like”.

The following edits are suggested:

L30: Evidence suggests that…to

L31: The ROS and antioxidants… in on cancer.

L32: The concentrations …exhibit(s) … necessitates necessity to understand..

L35: Present…This review…

L37: We conclude that by improving genetic screening methods and bringing higher clarity in determination of enzymatic pathways, and scale-up in cancer models profiling, using omics technology, would support in -depth understanding of antioxidant pathways and ROS complexities.

L45: The Cellular…function( s).

L47: levels...ROS levels …to..for

L53: ROS are considered as a diverse group of molecules that exert distinctive effects on cell components, over cellular processes that lead to pro-cancer and anti-cancer effects [2].

L61: cells…the…

L65: So Thus, this review focuses on …the…

L68: The ROS contain a minimum of one oxygen. Previous studies have reported…

L84: cause….exert…

L85: protein (s)

L89: The majority….cell damage…

L95: which that

L97: cell death…Figure 1 represents…

L101: Radiation is…a..for..of ROS

L104: and the ROS..

L106: allows..

L110: ROS…

L115: tumors…depend(s)

L118: Excessively high levels of ROS

L121: triggers….factor…

L122: Reports suggest deactivation

L128: Investigations report…

L139: cofactors that eliminate…

L141: before ago)…

L146:utilize as a cofactor..

L147-8: as a cofactor…comprise..

L149: exhibit…

L167: carcinogens or loss of tumor suppressors (loss).

L169: tumors…

L170: exhibit…

L171: One study ( reference) suggests…

L173: system is (are) Reports high light…Loss of PRDXI causes..

L176: Apart from this. It…

L178: exhibit( s)

L180: A high…

L181: repairs…lung tumors..

L182: (over) , of

L184: an important..

L186: tumors…

L188: a nutshell…

L200: during

L205: the

L206: support..,

L207: the

L208: The, a, tumors..

L210: lung(s)…

L212: as a challenge…

L265: involved in the tumorigenic…

L294: leads…

L297: models

L306: exhibit(s) …functions

L320-1: probes…undergo…

L332: not the same…

L342: which is necessary for…

L343: systems…

L345: necessary…must

L358: Like…To detect

L359: In vitro incubation of apocyanin is required…

L384: It presents a broad ….

L393: cell membranes…

L396: enzyme (s)

L412: a reduced…

L450: studies…

L469: exhibit…

L501: One study (   ) claims…

L515: Probes have been (are) …Like In order…

L517: that they assist in the…

L518: in (the) mammalian cells….assist (s)…

L519: offer (s)…

L522: confusing sentence; ( to oxygen or organ)?..is (are)..526:

L526: a… on a reversible…

L528: offer…

L530: Like…Use of…

L531: Sample (s)…

L532: mammalian cell analysis…

L533: tissues with high…

L534: with…at the cellular …

L568: the L012 does not react directly with…

L584: One study (   ) suggests…

L587: Liquid chromatography…

L590: The Chromatography

L595: react (s)

L605: the

L614: One study (181) suggests a relationship….

L629: to

L631: The NBT is another often (most)…

L650: generates.. generating an single…

L656: make the boronate…

L661: when it reacts with…

L662: a brown…

L688: analogue (s)…

L693: makes making..

L701: exhibit (s)…

L702: in using…use of…

L704: a set of…

L707: affect ( s)

L708: temperature affects…

L709: radical (s)…a liquid…

L712: systems…

L713: Any aforesaid…..

L715: The majority..

L728: probes (208)…

L735: The Table 1…as well as..

L741: the future.. Current “state of the art” endorses..

L742: differ (s)..

L744: the same cells…

L749: This..It

L750: relationship.. Present studies recommend (s)

L754: This review highlights… such as (like)…

L756: The current review…

L759: Study establishes that…Combination of

L767: The present review recommends…

L768: but….

Author Response

                                                                                                                                                                                                                    Date: 10-01-2021

Cover Letter

To,

The Chief Editor,

Antioxidants-MDPI

 Subject: Submission of revised manuscript (id: antioxidants-1054926)

Dear  editor, 

Thanks to your editorial team and reviewers for the valuable comments and suggestions to improve the quality of our manuscript Dear Chief Editor,

titled “Comprehensive review of methodology to detect reactive oxygen species (ROS) in mammalian species and establish its relationship with antioxidants and cancer.

Based on reviewer comments the manuscript has been revised/modified.

Please find below the point by point details of the revision in the manuscript.

Thank and Regards,

Dr. Neeraj Kumar Fuloria

Snr. Associate Professor

Faculty of Pharmacy

AIMST University

Kedah, Bedong 08100, Malaysia

Email: neerajkumar@aimst.edu.my

POINT BY POINT DETAILS OF THE REVISION IN THE MANUSCRIPT

REVIEWERS COMMENTS AND JUSTIFCATION

Authors are sincerely thankful to the reviewer’s comments, as modification of manuscript based on reviewers comment will certainly enhance the quality of manuscript and citations.

REVIEWER 1

Comments:

  • This review provides considerable valuable intellectual perspective on a complex topic. There are many references on antioxidant functions and responses in mammalian tissues. While there may be some cross-application, review of plant systems are essentially non-existent. One suggestion may be to indicate this review is primarily for mammalian free radicals and antioxidants. For example, change the title:

Comprehensive review of methodology to detect reactive oxygen species (ROS) in mammalian species and establish its relationship with antioxidants and cancer

As per suggestion manuscript title is revised as “Comprehensive review of methodology to detect reactive oxygen species (ROS) in mammalian species and establish its relationship with antioxidants and cancer”.

  • The graphics are very good and useful to assist following the review comments. In some instances clarity may be improved by reducing wordiness of the text.

Authors appreciates reviewer comments. As per the suggestions figure configuration has been modified.

  • Some observations and comments suggest improvements in: adherence to agreement between singular/plural structure; choice of ‘article’ or no use of article; elimination and or choice of unnecessary modifying phrases, avoid use of “Like”.
  • The following edits are suggested:
  • L30: Evidence suggests that…to
  • L31: TheROS and antioxidants… in on cancer.
  • L32: The concentrations …exhibit(s) … necessitatesnecessity to understand..
  • L35: Present…This review…
  • L37: We conclude that by improving genetic screening methods and bringing higher clarity in determination of enzymatic pathways, and scale-up in cancer models profiling, using omics technology, would support in -depth understanding of antioxidant pathways and ROS complexities.
  • L45: TheCellular…function( s).
  • L47: levels...ROS levels …to..for
  • L53: ROS are considered as a diverse group of molecules that exert distinctive effects on cell components, over cellular processes that lead to pro-cancer and anti-cancer effects [2].
  • L61: cells…the…
  • L65: So Thus, this review focuses on …the…
  • L68: TheROS contain a minimum of one oxygen. Previous studies have reported…
  • L84: cause….exert…
  • L85: protein (s)
  • L89: The majority….cell damage…
  • L95: whichthat
  • L97: cell death…Figure 1 represents…
  • L101: Radiation is…a..for..of ROS
  • L104: and the ROS..
  • L106: allows..
  • L110: ROS…
  • L115: tumors…depend(s)
  • L118: Excessively high levels of ROS
  • L121: triggers….factor…
  • L122: Reports suggest deactivation
  • L128: Investigations report…
  • L139: cofactors that eliminate…
  • L141: before ago)…
  • L146:utilize as a cofactor..
  • L147-8: as a cofactor…comprise..
  • L149: exhibit…
  • L167: carcinogens or loss of tumor suppressors (loss).
  • L169: tumors…
  • L170: exhibit…
  • L171: One study ( reference) suggests…
  • L173: system is (areReports high light…Loss of PRDXI causes..
  • L176: Apart from this. It…
  • L178: exhibit( s)
  • L180: A high…
  • L181: repairs…lung tumors..
  • L182: (over) , of
  • L184: an important..
  • L186: tumors…
  • L188: a nutshell…
  • L200: during
  • L205: the
  • L206: support..,
  • L207: the
  • L208: The, a, tumors..
  • L210: lung(s)…
  • L212: as a challenge…
  • L265: involved in the tumorigenic…
  • L294: leads…
  • L297: models
  • L306: exhibit(s) …functions
  • L320-1: probes…undergo…
  • L332: not the same…
  • L342: which is necessary for…
  • L343: systems…
  • L345: necessary…must
  • L358: Like…To detect
  • L359: In vitro incubation of apocyanin is required…
  • L384: It presents a broad ….
  • L393: cell membranes…
  • L396: enzyme (s)
  • L412: a reduced…
  • L450: studies…
  • L469: exhibit…
  • L501: One study (   ) claims…
  • L515: Probes have been (are) …LikeIn order…
  • L517:that they assist in the…
  • L518: in (the) mammalian cells….assist (s)…
  • L519: offer (s)…
  • L522: confusing sentence; ( to oxygen or organ)?..is (are)..526:
  • L526: a… on a reversible…
  • L528: offer…
  • L530: Like…Use of…
  • L531: Sample (s)…
  • L532: mammalian cell analysis…
  • L533: tissues with high…
  • L534: with…at the cellular …
  • L568: the L012 does not react directly with…
  • L584: One study (   ) suggests…
  • L587: Liquid chromatography…
  • L590: TheChromatography
  • L595: react (s)
  • L605: the
  • L614: One study (181) suggests a relationship….
  • L629: to
  • L631: TheNBT is another often (most)…
  • L650: generates.. generating an single…
  • L656: make the boronate…
  • L661: when it reacts with…
  • L662: a brown…
  • L688: analogue (s)…
  • L693: makes.
  • L701: exhibit (s)…
  • L702: in using…use of…
  • L704: a set of…
  • L707: affect ( s)
  • L708: temperature affects…
  • L709: radical (s)…a liquid…
  • L712: systems…
  • L713: Any aforesaid…..
  • L715: The majority..
  • L728: probes (208)…
  • L735: TheTable 1…as well as..
  • L741: the future.. Current “state of the art” endorses..
  • L742: differ (s)..
  • L744: the same cells…
  • L749: This..It
  • L750:Present studies recommend (s)
  • L754: This review highlights… such as (like)…
  • L756: The current review…
  • L759: Study establishes that…Combination of
  • L767: The present review recommends…
  • L768: but….

Authors are sincerely thankful to the reviewer for the critical analysis of manuscript. As modifications based on suggestion/comments will definitely improve the quality of our manuscript.

As per the suggestion/comments the manuscript has been revised and highlighted with yellow colour.

Reviewer 2 Report

This review describes the origin of ROS, their effects on tumor development, the different types of antioxidants and the different methods to detect ROS.

This review is quite complete even if at times it is quite difficult to read.

However, in endogenous antioxidants section, the description of catalase and SOD1 and SOD2 is missing.

In the exogenous antioxidants section, the antioxidants brought by the diet are missing: ß-carotene, vitamin c, vitamin E, polyphenols and lycopenes.

Line 84-86: the same dose of 100 nM is mentioned for low and high doses of H2O2.

Author Response

                                                                                                                                                                                                                   Date: 10-01-2021

Cover Letter

To,

The Chief Editor,

Antioxidants-MDPI

 Subject: Submission of revised manuscript (id: antioxidants-1054926)

Dear editor, 

Thanks to your editorial team and reviewers for the valuable comments and suggestions to improve the quality of our manuscript Dear Chief Editor,

titled “Comprehensive review of methodology to detect reactive oxygen species (ROS) in mammalian species and establish its relationship with antioxidants and cancer.

Based on reviewer comments the manuscript has been revised/modified.

Please find below the point by point details of the revision in the manuscript.

Thank and Regards,

Dr. Neeraj Kumar Fuloria

Snr. Associate Professor

Faculty of Pharmacy

AIMST University

Kedah, Bedong 08100, Malaysia

Email: neerajkumar@aimst.edu.my

POINT BY POINT DETAILS OF THE REVISION IN THE MANUSCRIPT

REVIEWERS COMMENTS AND JUSTIFCATION

Authors are sincerely thankful to the reviewer’s comments, as modification of manuscript based on reviewers comment will certainly enhance the quality of manuscript and citations.

REVIEWER 2

Comments:

  • This review describes the origin of ROS, their effects on tumor development, the different types of antioxidants and the different methods to detect ROS.

Authors appreciate reviewers’ motivational words

  • This review is quite complete even if at times it is quite difficult to read.

Authors appreciate reviewers’ kind words

  • However, in endogenous antioxidants section, the description of catalase and SOD1 and SOD2 is missing.

As per the reviewer comments the description of catalase and SOD1 and SOD2 has been incorporated and highlighted with sky blue colour.

  • In the exogenous antioxidants section, the antioxidants brought by the diet are missing: ß-carotene, vitamin c, vitamin E, polyphenols and lycopenes.

As required a detailed description is included in the exogenous antioxidants section and highlighted with sky blue colour.

  • Line 84-86: the same dose of 100 nM is mentioned for low and high doses of H2O2.

It was a typo error, the lower dose is 1-10 nm the same is now modified in the manuscript and highlighted with sky blue colour.

Reviewer 3 Report

General comments: This is an impressive and important presentation of the role of ROS for oncogenesis and its control in part 1 of the manuscript, as well as a very careful presentation and critical discussion of methods to detect specific ROS in the second part. The second part has been written in a very precise and comprehensive way. Therefore it will be of great benefit for scientists working in this field. I have no suggestions for further improvement of part 2.

Due the extremely large amount of literature on ROS and oncogenesis, it obviously cannot be avoided to focus on specific aspects and possibly neglect others in part 1 of the manuscript. Therefore, for further improvement of the manuscript I will make some suggestions for inclusion of additional aspects that are not addressed so far.

Specific comments:

1) line 71 (superoxide anions): there is a large body of literature on superoxide anion production and its role for induction of proliferation of malignant cells, as well as the role of superoxide anions for the induction of specific intercellular signaling pathways that endanger malignant cells. It seems to be important to me to include extracellular superoxide anion production through membrane-associated NADPH oxidase in malignant cells (references: Irani K et al., Science 1997; 275:1649-1652; Irani K and Goldschmidt-Clermont PJ: Biochem. Pharmacol. 1998; 55:1339-1346; reviewed in Bauer G. Anticancer Res. 2014; 34: 1467-1482).

2) line 76 ff: please mention also the interaction between extracellular superoxide anions and NO, leading to peroxynitrite. This reaction attacks particularly malignant cells with active NADPH oxidase (Heigold S, et al, Carcinogenesis 2002; 23: 929-941.). Peroxynitrite may either react with CO2, or if it is generated in close vicinity to proton pumps, it may get protonated to peroxynitrous acid. Peroxynitrous acid then decomposes into NO2 and hydroxyl radicals. These are damaging and also can induce apoptosis through lipid peroxidation

3) H2O2 does not diffuse through membranes, but is transported through aquaporins (Bienert GP et al.,  Biochem. Biophys. Acta. 2006; 1758: 994-1003).

4) line 79 ff: please mention that H2O2 is also the substrate for HOCl synthesis by MPO, peroxidasins or DUOX. HOCl undergoes Fenton chemistry much more efficiently than H2O2 (yielding hydroxyl radicals and chloride). Furthermore, the interaction between HOCl and superoxide anions leads to the formation of hydroxyl radicals. This reaction seems to be important for the elimination of malignant cells (reviewed in: Bauer G. J. Inorganic. Biochem. 2018; 179:10-23.  doi:10.1016/j.jinorgbio.2017.11.005).

5) line 88: please change peroxyl radical to hydroxl radical

6) Figure 1: the figure shows membrane-associated NADPH oxidase, Please refer in the text to the generation of superoxide anions by malignant cells through NADPH activity.

7) line 117: please include the selective ROS/RNS-dependent antitumor effects and the protection of cells from late stage of tumorigenesis (bona fide tumor cells) to protect themselves towards ROS/RNS signaling through expression of membrane-associated catalase; as well as efforts to use ROS signaling for tumor therapy. You can find relevant references and concepts in: Bauer G et al.,  Sci. Rep. 2019; 9: 14210; https://doi.org/10.1038/s41598-019-50291-0 and Bauer G.  Redox Biol. 2019; 26: 101291. https://doi.org/10.1016/j.redox.2019.101291 )

8) Figure 2: The Xc transporter transports glutamate out of cells and cystine (not cysteine) into the cells. Cystine is then converted to two molecules of cysteine by NADPH, and then glutathions synthesis begins.  Please correct.

Author Response

                                                                                                                                                                                                                      Date: 10-01-2021

Cover Letter

To,

The Chief Editor,

Antioxidants-MDPI

 Subject: Submission of revised manuscript (id: antioxidants-1054926)

 Dear editor,

Thanks to your editorial team and reviewers for the valuable comments and suggestions to improve the quality of our manuscript Dear Chief Editor,

titled “Comprehensive review of methodology to detect reactive oxygen species (ROS) in mammalian species and establish its relationship with antioxidants and cancer.

Based on reviewer comments the manuscript has been revised/modified.

Please find below the point by point details of the revision in the manuscript.

Thank and Regards,

Dr. Neeraj Kumar Fuloria

Snr. Associate Professor

Faculty of Pharmacy

AIMST University

Kedah, Bedong 08100, Malaysia

Email: neerajkumar@aimst.edu.my

POINT BY POINT DETAILS OF THE REVISION IN THE MANUSCRIPT

REVIEWERS COMMENTS AND JUSTIFCATION

Authors are sincerely thankful to the reviewer’s comments, as modification of manuscript based on reviewers comment will certainly enhance the quality of manuscript and citations.

REVIEWER 3

General Comments:

  • This is an impressive and important presentation of the role of ROS for oncogenesis and its control in part 1 of the manuscript, as well as a very careful presentation and critical discussion of methods to detect specific ROS in the second part. The second part has been written in a very precise and comprehensive way. Therefore it will be of great benefit for scientists working in this field. I have no suggestions for further improvement of part 2.

Authors appreciates the motivational words of reviewer.

  • Due the extremely large amount of literature on ROS and oncogenesis, it obviously cannot be avoided to focus on specific aspects and possibly neglect others in part 1 of the manuscript. Therefore, for further improvement of the manuscript I will make some suggestions for inclusion of additional aspects that are not addressed so far.

Authors appreciates reviewer suggestion as modifications as per the reviewer suggestions will definitely enhance the quality of manuscript.

Specific Comments:

  • 1) line 71 (superoxide anions): there is a large body of literature on superoxide anion production and its role for induction of proliferation of malignant cells, as well as the role of superoxide anions for the induction of specific intercellular signaling pathways that endanger malignant cells. It seems to be important to me to include extracellular superoxide anion production through membrane-associated NADPH oxidase in malignant cells (references: Irani K et al., Science 1997; 275:1649-1652; Irani K and Goldschmidt-Clermont PJ: Biochem. Pharmacol. 1998; 55:1339-1346; reviewed in Bauer G. Anticancer Res. 2014; 34: 1467-1482).

Authors appreciate reviewer suggestion/comments. The required information is incorporated into suggested section of manuscript and highlighted with green colour.

  • 2) line 76 ff: please mention also the interaction between extracellular superoxide anions and NO, leading to peroxynitrite. This reaction attacks particularly malignant cells with active NADPH oxidase (Heigold S, et al, Carcinogenesis 2002; 23: 929-941.). Peroxynitrite may either react with CO2, or if it is generated in close vicinity to proton pumps, it may get protonated to peroxynitrous acid. Peroxynitrous acid then decomposes into NO2 and hydroxyl radicals. These are damaging and also can induce apoptosis through lipid peroxidation

Authors are thankful to reviewer for the kind suggestion/information. The suggested information has been incorporated and highlighted with green colour.

  • 3) H2O2 does not diffuse through membranes, but is transported through aquaporins (Bienert GP et al.,  Biochem. Biophys. Acta. 2006; 1758: 994-1003).

The required information has been incorporated and highlighted with green colour.

  • 4) line 79 ff: please mention that H2O2 is also the substrate for HOCl synthesis by MPO, peroxidasins or DUOX. HOCl undergoes Fenton chemistry much more efficiently than H2O2 (yielding hydroxyl radicals and chloride). Furthermore, the interaction between HOCl and superoxide anions leads to the formation of hydroxyl radicals. This reaction seems to be important for the elimination of malignant cells (reviewed in: Bauer G. J. Inorganic. Biochem. 2018; 179:10-23.  doi:10.1016/j.jinorgbio.2017.11.005).

Authors are thankful to reviewer for the kind suggestion/information. The suggested information has been incorporated and highlighted with green colour.

  • 5) line 88: please change peroxyl radical to hydroxl radical

It was a typo error, the same has been corrected and highlighted with green colour.

  • 6) Figure 1: the figure shows membrane-associated NADPH oxidase, Please refer in the text to the generation of superoxide anions by malignant cells through NADPH activity.

As per the reviewer suggestion the required information has been incorporated in the text and highlighted with green colour.

  • 7) line 117: please include the selective ROS/RNS-dependent antitumor effects and the protection of cells from late stage of tumorigenesis (bona fide tumor cells) to protect themselves towards ROS/RNS signaling through expression of membrane-associated catalase; as well as efforts to use ROS signaling for tumor therapy. You can find relevant references and concepts in: Bauer G et al.,  Sci. Rep. 2019; 9: 14210; https://doi.org/10.1038/s41598-019-50291-0 and Bauer G.  Redox Biol. 2019; 26: 101291. https://doi.org/10.1016/j.redox.2019.101291)

As per the reviewer suggestion the required information has been incorporated in the text and highlighted with green colour.

  • 8) Figure 2: The Xc transporter transports glutamate out of cells and cystine (not cysteine) into the cells. Cystine is then converted to two molecules of cysteine by NADPH, and then glutathions synthesis begins.  Please correct.

As per the reviewer comment, figure 2 has been modified.

Round 2

Reviewer 1 Report

I am pleased the authors have taken the comments and recommendations seriously and have provided revisions that have improved readability of the manuscript.

Reviewer 2 Report

The authors have improved their manuscript. I congratulate them for the hard work done in this review.